# GeoClip: Geometry-Aware Clipping for Differentially Private SGD

**Atefeh Gilani**
Arizona State University
Tempe, AZ, USA
agilani2@asu.edu

**Naima Tasnim**
Arizona State University
Tempe, AZ, USA
ntasnim2@asu.edu

**Lalitha Sankar**
Arizona State University
Tempe, AZ, USA
lsankar@asu.edu

**Oliver Kosut**
Arizona State University
Tempe, AZ, USA
okosut@asu.edu

## Abstract

Differentially private stochastic gradient descent (DP-SGD) is the most widely used method for training machine learning models with provable privacy guarantees. A key challenge in DP-SGD is setting the per-sample gradient clipping threshold, which significantly affects the trade-off between privacy and utility. While recent adaptive methods improve performance by adjusting this threshold during training, they operate in the standard coordinate system and fail to account for correlations across the coordinates of the gradient. We propose GeoClip, a geometry-aware framework that clips and perturbs gradients in a transformed basis aligned with the geometry of the gradient distribution. GeoClip adaptively estimates this transformation using only previously released noisy gradients, incurring no additional privacy cost. We provide convergence guarantees for GeoClip and derive a closed-form solution for the optimal transformation that minimizes the amount of noise added while keeping the probability of gradient clipping under control. Experiments on both tabular and image datasets demonstrate that GeoClip consistently outperforms existing adaptive clipping methods under the same privacy budget.

## 1 Introduction

As machine learning models are increasingly trained on sensitive user data, ensuring strong privacy guarantees during training is essential to reduce the risk of misuse, discrimination, or unintended data exposure. Differential privacy (DP) [1, 2] offers a principled framework for protecting individual data, and has become a cornerstone of privacy-preserving machine learning. In deep learning, the most widely used approach for DP is differentially private stochastic gradient descent (DP-SGD) [3], which clips per-sample gradients and adds calibrated noise to their average.

Despite its widespread use, standard DP-SGD has a key limitation: it relies on a fixed clipping threshold to bound the sensitivity of individual gradients. Selecting this threshold poses a challenging privacy-utility tradeoff—setting it too low discards useful gradient information, while setting it too high increases sensitivity and necessitates injecting more noise, ultimately degrading model performance. This trade-off was observed empirically by McMahan et al. [4] and later analyzed theoretically and shown to be a fundamental limitation of differentially private learning by Amin et al. [5]. Moreover, the optimal threshold can vary over the course of training, across tasks, and between datasets, limiting the effectiveness of a fixed setting.

39th Conference on Neural Information Processing Systems (NeurIPS 2025).

To address this, recent work has proposed adaptive strategies that dynamically adjust the clipping threshold during training. One class of methods uses decay schedules to reduce the threshold over time. Yu et al. [6] and Du et al. [7] propose linear and near-linear decay rules, respectively, where the schedule is predefined and does not depend on the dataset—hence, no privacy budget is required. Lin et al. [8] introduce a nonlinear decay schedule, along with a transfer strategy that leverages public data to guide threshold selection. Recently, methods have been introduced which set the clipping level based on the data during the training process. These include AdaClip [9], which applies coordinate-wise clipping based on estimated gradient variances. Yet another is quantile-based clipping [10], which sets the threshold using differentially private quantiles of per-sample gradient norms. Although [10] was designed for federated learning, it can be adapted to centralized DP-SGD. These adaptive strategies have been shown to improve model utility while preserving privacy guarantees.

Despite these advances, existing adaptive clipping methods remain agnostic to the geometry of the gradient distribution. They operate in the standard basis—treating each coordinate independently. This overlooks dependencies between coordinates, especially when gradients exhibit strong correlations across dimensions. In such cases, independently clipping and perturbing each coordinate can introduce redundant noise without improving privacy, ultimately degrading model utility. To address this, we propose *GeoClip*, a method that transforms gradients into a decorrelated basis that better reflects their underlying geometry. By applying DP mechanisms in this transformed space, GeoClip allocates noise more effectively, achieving a better privacy-utility tradeoff.

A different approach to correlations in gradients from the literature considers introducing correlations—dependencies across iterations and between entries of the noise vector—into the noise, rather than injecting i.i.d. Gaussian noise. This has been shown to improve the utility of private training [11, 12]. Choquette-Choo et al. [13] strengthen this direction by analytically characterizing near-optimal spatio-temporal correlation structures that lead to provably tighter privacy-utility tradeoffs. However, these approaches have been developed independently of adaptive clipping, and the effect of combining both methods remains unexplored. In addition, [13] uses pre-determined correlations in the noise, rather than being tailored to the data, as our approach is.

GeoClip is data-driven but does not require any additional privacy budget to compute the clipping transformation. Instead, it reuses the noisy gradients already released during training to estimate the mean and the covariance of the gradient distribution. By reusing these privatized gradients, GeoClip adapts its basis over time without accessing raw data or incurring additional privacy cost.

We list our main contributions below:

1. We propose GeoClip, a novel framework that applies differential privacy in a transformed basis rather than the standard coordinate system. To guide the choice of transformation, we derive a convergence theorem (Theorem 1) showing how the basis impacts convergence under DP-SGD, providing theoretical guidance for selecting transformations that improve utility.

2. Building on this insight, we formulate a convex optimization problem to find the transformation, and derive a closed-form solution (Theorem 2).

3. We introduce two algorithms to estimate the transformation using only previously released noisy gradients. The first is based on a moving average to estimate the gradient covariance matrix.

4. For large-scale models, the full gradient covariance matrix is prohibitively large to store. Thus, our second algorithm uses a streaming low-rank approximation of the covariance matrix. This second algorithm is thus suitable for deep models with large parameter counts.

5. We validate GeoClip through experiments on synthetic, tabular, and image datasets, showing that it consistently outperforms existing adaptive clipping methods under the same privacy budget.

**Notation.** We denote the $d \times d$ identity matrix by $I_d$. The notation $N \sim \mathcal{N}(0, \sigma^2 I_d)$ denotes a $d$-dimensional Gaussian with zero mean and covariance $\sigma^2 I_d$. We use $\|x\|_2$ for the Euclidean norm of a vector $x$, $A^{-1}$ for the inverse of matrix $A$, and $A^\top$ for its transpose. The trace of a matrix $A$ is denoted by $\mathrm{Tr}(A)$, and $\mathrm{Cov}(x \mid y)$ refers to the conditional covariance of $x$ given $y$.

## 2 General Framework

Let $\mathcal{D} = \{x_k\}_{k=1}^N$ be a dataset of $N$ examples, and let $f : \mathbb{R}^d \to \mathbb{R}$ denote the empirical loss function defined as the average of per-sample losses:

$$f(\theta) = \frac{1}{N} \sum_{k=1}^N f_k(\theta), \tag{1}$$

where $\theta \in \mathbb{R}^d$ is the model parameter vector and each $f_k$ corresponds to the loss on the $k$-th data point $x_k$. In DP-SGD, the algorithm updates $\theta$ using a noisy clipped stochastic gradient to ensure privacy. Let $g_t \in \mathbb{R}^d$ denote the stochastic gradient at iteration $t$. The update rule is:

$$\theta_{t+1} = \theta_t - \eta \tilde{g}_t, \tag{2}$$

where $\tilde{g}_t$ is the privatized version of $g_t$, obtained by clipping and adding noise.

GeoClip builds on the DP-SGD framework but changes how gradients are processed before clipping and adding noise. It begins by shifting and projecting the gradient into a new coordinate system:

$$\omega_t = M_t(g_t - a_t), \tag{3}$$

where $a_t \in \mathbb{R}^d$ is a reference point and $M_t \in \mathbb{R}^{d \times d}$ is a full-rank transformation matrix that defines the new basis. To enforce differential privacy, we clip the transformed gradient $\omega_t$ to unit norm and add Gaussian noise:

$$\tilde{\omega}_t = \frac{\omega_t}{\max(1, \|\omega_t\|_2)} + N_t, \quad N_t \sim \mathcal{N}(0, \sigma^2 I_d), \tag{4}$$

where $\sigma$ is set based on the desired privacy guarantee. We then map the noisy, clipped gradient back to the original space:

$$\tilde{g}_t = M_t^{-1} \tilde{\omega}_t + a_t. \tag{5}$$

**Remark 1.** *GeoClip generalizes AdaClip of Pichapati et al. [9], which itself extends standard DP-SGD. AdaClip essentially assumes that $M_t$ is diagonal for per-coordinate scaling, whereas GeoClip allows $M_t$ to be any full-rank matrix. This added flexibility allows GeoClip to account for correlations between gradient components and inject noise along more meaningful directions.*

A key feature of our framework is that the privacy guarantees remain unaffected by the choice of $M_t$ and $a_t$, as long as no privacy budget is used to compute them. This allows $M_t$ and $a_t$ to be chosen entirely based on utility, without compromising privacy. The main challenge, then, is how to select these parameters effectively. To address this, we first define a performance metric for GeoClip. Inspired by Pichapati et al. [9], we derive the following convergence bound for our framework and use it to guide the design of $M_t$ and $a_t$, ultimately improving both convergence and training efficiency.

**Theorem 1** (Convergence of GeoClip). *Assume $f$ has an $L$-Lipschitz continuous gradient. Further, assume the stochastic gradients are bounded, i.e., $\|\nabla f_k(\theta)\| \leq G$, and have bounded variance, i.e., $\mathbb{E}_k \|\nabla f_k(\theta) - \nabla f(\theta)\|^2 \leq \sigma_g^2$. Let $\theta^* = \arg\min_{\theta \in \mathbb{R}^d} f(\theta)$ denote the optimal solution, and suppose the learning rate satisfies $\eta < \frac{2}{3L}$. Then, for the iterates $\{\theta_t\}_{t=0}^{T-1}$ produced by GeoClip with batch size 1 using the update rule $\theta_{t+1} = \theta_t - \eta \tilde{g}_t$, where $\tilde{g}_t$ is defined in (5), the average squared gradient norm satisfies:*

$$\frac{1}{T} \sum_{t=0}^{T-1} \mathbb{E}\|\nabla f(\theta_t)\|^2 \leq \underbrace{\frac{f(\theta_0) - f(\theta^*)}{T\left(\eta - \frac{3L\eta^2}{2}\right)}}_{\text{Optimization gap}} + \underbrace{\frac{3L\eta}{2 - 3L\eta}\sigma_g^2}_{\text{Gradient variance term}} + \underbrace{\frac{L\eta\sigma^2}{T(2 - 3L\eta)} \sum_{t=0}^{T-1} \mathbb{E} \operatorname{Tr}\left[\left(M_t^\top M_t\right)^{-1}\right]}_{\text{Noise-injection term}}$$

$$+ \underbrace{\frac{2}{T(2 - 3L\eta)} \sum_{t=0}^{T-1} \mathbb{E}\left[\beta(a_t)\left(\operatorname{Tr}\left(M_t^\top M_t \Sigma_t\right) + \|M_t(\mathbb{E}[g_t \mid \theta^t] - a_t)\|^2\right)\right]}_{\text{Clipping error term}}, \tag{6}$$

where $\theta^t = (\theta_0, \ldots, \theta_t)$ represents the history of parameter values up to iteration $t$, $\Sigma_t = \mathrm{Cov}(g_t | \theta^t)$, and

$$\beta(a_t) = (G + \|a_t\|) \left( G + \frac{3L\eta}{2}(G + \|a_t\|) \right). \tag{7}$$

The above result generalizes Theorem 2 in Pichapati et al. [9]; proof details are in Appendix A.

Theorem 1 provides insight into how we should choose the transformation parameters $a_t$ and $M_t$. In particular, we want to choose the transformation parameters $a_t$ and $M_t$, for all $t$, to minimize the right-hand side of (6). The reference point $a_t$ directly affects the clipping error by setting the center around which gradients are clipped, thereby influencing how much of each gradient is truncated. In contrast, the noise injection term remains independent of $a_t$, as noise is added regardless of the gradient's position relative to $a_t$. From Theorem 1, the clipping error at iteration $t$ takes the form

$$\beta(a_t) \left( \mathrm{Tr}\left( M_t^\top M_t \Sigma_t \right) + \|M_t(\mathbb{E}[g_t \mid \theta^t] - a_t)\|^2 \right), \tag{8}$$

where $\beta(a_t)$ is a scale factor that grows with $\|a_t\|$. A natural choice is to set $a_t = \mathbb{E}[g_t \mid \theta^t]$ which eliminates the bias term $\|M_t(\mathbb{E}[g_t \mid \theta^t] - a_t)\|^2$. Given this choice, since gradients are norm-bounded by $G$, we can apply Jensen's inequality to also bound the norm of their mean, to obtain the bound

$$\beta(a_t) \leq 2G^2(1 + 3L\eta). \tag{9}$$

We now show that the remaining contribution to the clipping error term $\mathrm{Tr}\left( M_t^\top M_t \Sigma_t \right)$ in fact serves as an upper bound on the probability that the gradient is clipped (i.e., $\|\omega\| > 1$). We do so using Markov's inequality as detailed below:

$$\mathrm{Tr}\left( M_t^\top M_t \Sigma_t \right) = \mathbb{E}[\|M_t(g_t - \mathbb{E}[g_t \mid \theta^t])\|^2 \mid \theta^t] \tag{10}$$

$$= \mathbb{E}[\|\omega\|^2 \mid \theta^t] \tag{11}$$

$$\geq \Pr(\|\omega\| > 1 \mid \theta^t). \tag{12}$$

We can now interpret the clipping error term as the likelihood that clipping occurs—ideally, the lower the better as the gradient information will be better preserved. However, in setting a clipping level, we must also be aware of the amount of noise: if $M_t$ is scaled down, there is effectively more noise, as captured by the noise-injection term in (6). We handle this tradeoff via the following optimization problem for the transformation matrix $M_t$:

$$\underset{M_t}{\text{minimize}} \quad \mathrm{Tr}\left( M_t^\top M_t \right)^{-1}$$

$$\text{subject to} \quad \mathrm{Tr}\left( M_t^\top M_t \Sigma_t \right) \leq \gamma. \tag{13}$$

**Theorem 2.** *Let $\Sigma_t = \mathrm{Cov}(g_t \mid \theta^t)$ be a positive definite matrix. The optimal transformation matrix $M_t^* \in \mathbb{R}^{d \times d}$ at iteration $t$ for the optimization problem in (13), along with its corresponding objective value, are given by:*

$$M_t^* = \left( \frac{\gamma}{\sum_{i=1}^d \sqrt{\lambda_i}} \right)^{1/2} \Lambda_t^{-1/4} U_t^\top, \quad \mathrm{Tr}\left( M_t^{*\top} M_t^* \right)^{-1} = \frac{\left( \sum_{i=1}^d \sqrt{\lambda_i} \right)^2}{\gamma}, \tag{14}$$

*where $\Sigma_t = U_t \Lambda_t U_t^\top$ is the eigendecomposition of the covariance matrix, with $\Lambda_t = \mathrm{diag}(\lambda_1, \ldots, \lambda_d)$ containing its eigenvalues.*

**Remark 2.** *Applying the optimal transformation $M_t^*$ to the gradient $g_t$ at iteration $t$, the covariance of the transformed gradient $\tilde{\omega}_t = M_t^*(g_t - \mathbb{E}[g_t \mid \theta^t])$, conditioned on the history $\theta^t$, is*

$$Cov(\tilde{\omega}_t \mid \theta^t) = M_t^* \Sigma_t M_t^{*\top} = \frac{\gamma \, \Lambda_t^{1/2}}{\sum_{i=1}^d \sqrt{\lambda_i}} = \gamma \, \mathrm{diag}\left( \frac{\sqrt{\lambda_1}}{\sum_{i=1}^d \sqrt{\lambda_i}}, \ldots, \frac{\sqrt{\lambda_d}}{\sum_{i=1}^d \sqrt{\lambda_i}} \right).$$

*Thus, the optimal transformation decorrelates and scales down the gradients while preserving the relative ordering of variance across directions, in contrast to traditional whitening, which eliminates all variance structure. Note that setting $a_t = \mathbb{E}[g_t \mid \theta^t]$ and choosing $M_t = \sqrt{\frac{\gamma}{d}} \Lambda_t^{-1/2} U_t^\top$*

*would amount to a whitening of the gradients and ensures that the constraint in* (13) *is active, i.e.,* $\text{Tr}\left(M_t^\top M_t \Sigma_t\right) = \gamma$. *Under this choice, the objective becomes:*

$$\text{Tr}\left(M_t^\top M_t\right)^{-1} = \frac{d}{\gamma}\text{Tr}(\Lambda_t) = \frac{d}{\gamma}\sum_{i=1}^{d}\lambda_i. \tag{15}$$

*Comparing our objective in* (14) *with the objective resulting from the whitening transformation, we observe that applying the Cauchy–Schwarz inequality yields:*

$$\frac{d}{\gamma}\sum_{i=1}^{d}\lambda_i \geq \frac{1}{\gamma}\left(\sum_{i=1}^{d}\sqrt{\lambda_i}\right)^2, \tag{16}$$

*with equality if and only if* $\lambda_1 = \ldots = \lambda_d$. *This shows that our solution achieves a strictly smaller objective than whitening in all non-isotropic cases, where the gradient distribution exhibits unequal variance along the eigenbasis directions.*

The proof of Theorem 2 is provided in Appendix B.

## 3 Algorithm Overview

Algorithm 1 outlines our proposed GeoClip method. We explain its key steps below.

**Moving average for mean and covariance estimation.** While the theoretical results from Section 2 assume access to the true gradient distribution, in practice this distribution is unknown, and so it must be estimated using only privatized gradients. We estimate the mean and covariance using exponential moving averages computed from those privatized gradients. This enables us to estimate the geometry of the gradients without consuming additional privacy budget. Specifically, we maintain estimates $a_t$ of the mean, and $\Sigma_t$ of the covariance matrix, which are updated according to:

$$a_{t+1} \leftarrow \beta_1 a_t + (1 - \beta_1)\tilde{g}_t \tag{17}$$
$$\Sigma_{t+1} \leftarrow \beta_2 \Sigma_t + (1 - \beta_2)(\tilde{g}_t - a_t)(\tilde{g}_t - a_t)^\top \tag{18}$$

where $\beta_1$ and $\beta_2$ are constants close to 1 (e.g., $\beta_1 = 0.99$, $\beta_2 = 0.999$). The eigenvalues and eigenvectors used for the transformation are then computed from the estimated covariance.

**Clamping eigenvalues.** The covariance matrix is positive semi-definite and may contain zero eigenvalues, which can cause numerical instability. To address this, we clamp eigenvalues from below at a small threshold $h_1$ (e.g., $10^{-15}$). Since we only observe privatized gradients, which may be noisy and unstable, we also clamp from above at $h_2$ to prevent extreme scaling.

**Covariance update with mini-batch.** Let $B$ denote a mini-batch of training examples sampled at each iteration, with $|B|$ indicating the batch size. When using mini-batch gradient descent with $|B| > 1$, we must estimate the mean and covariance from the privatized batch averages of the per-sample gradients. Let $g_i$ be the random variable representing the gradient of the $i$-th sample in the batch. Let $\bar{g} = \frac{1}{|B|}\sum_{i \in B} g_i$ be the batch average gradient. Assuming that the $g_i$ are i.i.d., with the same distribution as $g$, the covariance of the average gradient $\bar{g}$ satisfies

$$\text{Cov}\left(\bar{g}\right) = \text{Cov}\left(\frac{1}{|B|}\sum_{i \in B} g_i\right) = \frac{1}{|B|}\text{Cov}(g). \tag{19}$$

The same principle applies when we observe only the privatized average gradient $\tilde{g}$. To account for this averaging effect, the covariance update is scaled by the batch size; i.e., line 11 of GeoClip in Algorithm 1 becomes

$$\Sigma_{t+1} \leftarrow \beta_2 \Sigma_t + |B|(1 - \beta_2)(\tilde{g}_t - a_t)(\tilde{g}_t - a_t)^\top. \tag{20}$$

**Low-Rank PCA.** When the dimensionality is high, computing and storing the full gradient covariance matrix becomes impractical. To address this, we propose a method to maintain a low-rank approximation using a simple and efficient procedure we refer to as *Streaming Rank-k PCA* (Algorithm 2). Specifically, we maintain an approximate eigendecomposition of the covariance in the form $U_t\Lambda_t U_t^\top$,

---

**Algorithm 1** GeoClip

---

**Require:** Dataset $\mathcal{D}$, model $f_\theta$, loss $\mathcal{L}$, learning rate $\eta$, noise scale $\sigma$, steps $T$, hyperparameters $h_1, h_2, \beta_1, \beta_2$

1: Initialize $\theta$, mean vector $a_0 = 0$, covariance $\Sigma_0 = I_d$, transform $M_0 = M_0^{\text{inv}} = I_d$
2: **for** $t = 0$ to $T$ **do**
3:     Sample a data point $(x_t, y_t)$
4:     Compute gradient $g_t \leftarrow \nabla_\theta \mathcal{L}(f_\theta(x_t), y_t)$
5:     Center and transform: $\omega_t \leftarrow M_t(g_t - a_t)$
6:     Clip: $\bar{\omega}_t \leftarrow \omega_t / \max(1, \|\omega_t\|_2)$
7:     Add noise: $\tilde{\omega}_t \leftarrow \bar{\omega}_t + N$, where $N \sim \mathcal{N}(0, \sigma^2 I_d)$
8:     Map back: $\tilde{g}_t \leftarrow M_t^{\text{inv}} \tilde{\omega}_t + a_t$
9:     Update model: $\theta_{t+1} \leftarrow \theta_t - \eta \tilde{g}_t$
10:    Update mean: $a_{t+1} \leftarrow \beta_1 a_t + (1 - \beta_1) \tilde{g}_t$
11:    Update covariance: $\Sigma_{t+1} \leftarrow \beta_2 \Sigma_t + (1 - \beta_2)(\tilde{g}_t - a_t)(\tilde{g}_t - a_t)^\top$
12:    Eigendecompose: $\Sigma_{t+1} = U_t \Lambda_t U_t^\top$
13:    Clamp eigenvalues: $\lambda_i \leftarrow \texttt{Clamp}(\lambda_i, \min = h_1, \max = h_2)$
14:    Set $M_{t+1} \leftarrow \left(\gamma / \sum_i \sqrt{\lambda_i}\right)^{1/2} \Lambda_t^{-1/4} U_t^\top$
15:    Set $M_{t+1}^{\text{inv}} \leftarrow \left(\gamma / \sum_i \sqrt{\lambda_i}\right)^{-1/2} U_t \Lambda_t^{1/4}$
16: **end for**
17: **return** Final parameters $\theta$

---

---

**Algorithm 2** STREAMING RANK-$k$ PCA

---

**Require:** Eigenvectors $U_t \in \mathbb{R}^{d \times k}$, eigenvalues $\Lambda_t \in \mathbb{R}^{k \times k}$, gradient $\tilde{g}_t \in \mathbb{R}^d$, mean $a_{t+1} \in \mathbb{R}^d$, factor $\beta_3 \in \mathbb{R}$, rank $k$

1: Center: $z \leftarrow \tilde{g}_t - a_{t+1}$
2: Form augmented matrix: $U_{\text{aug}} \leftarrow [U_t \ \ z]$
3: Compute: $Z \leftarrow U_{\text{aug}} \, \text{diag}(\sqrt{\beta_3 \lambda_1}, \ldots, \sqrt{\beta_3 \lambda_k}, \sqrt{1 - \beta_3})$
4: Perform SVD: $Z = VSR^\top$
5: Set $U_{t+1} \leftarrow$ first $k$ columns of $V$
6: Set $\Lambda_{t+1} \leftarrow$ squares of the first $k$ singular values in $S$
7: Return: $U_{t+1}, \Lambda_{t+1}$

---

where $U_t \in \mathbb{R}^{d \times k}$ contains the top-$k$ eigenvectors and $\Lambda_t \in \mathbb{R}^{k \times k}$ the corresponding eigenvalues, where $k \ll d$. Upon receiving a new gradient $\tilde{g}_t$, we center it using the running mean $a_{t+1}$, yielding $z = \tilde{g}_t - a_{t+1}$, and perform a weighted update to the covariance:

$$\Sigma_{t+1} = \beta_3 U_t \Lambda_t U_t^\top + (1 - \beta_3) z z^\top = [U_t \ \ z] \begin{bmatrix} \beta_3 \Lambda_t & 0 \\ 0 & 1 - \beta_3 \end{bmatrix} [U_t \ \ z]^\top \tag{21}$$

Rather than forming this full matrix, we compute its square root:

$$Z = [U_t \ \ z] \cdot \text{diag}(\sqrt{\beta_3 \lambda_1}, \ldots, \sqrt{\beta_3 \lambda_k}, \sqrt{1 - \beta_3}) \in \mathbb{R}^{d \times (k+1)}. \tag{22}$$

We then perform an SVD on $Z$ and retain the top $k$ singular vectors and squared singular values as the updated eigenvectors and eigenvalues. We note that this computation takes $\mathcal{O}(dk^2 + k^3)$ time [14], highlighting that it is only linear in $d$. The rest of the procedure follows Algorithm 1 by replacing the moving average over the full covariance matrix in line 11 of Algorithm 1 with the low-rank approximation described above. Another necessary change to the algorithm is the following: since $U_t$ is no longer a square matrix, the transformation $M_t$ takes the gradient into the lower $k$-dimensional space to clip and add noise, such that the resulting $M_t^{\text{inv}}$ returns to the full $d$-dimensional space. The complete version of this variant is provided as Algorithm 3 in Appendix C.

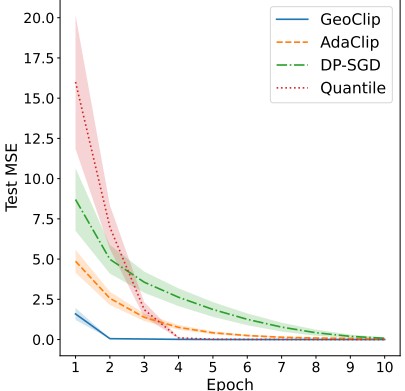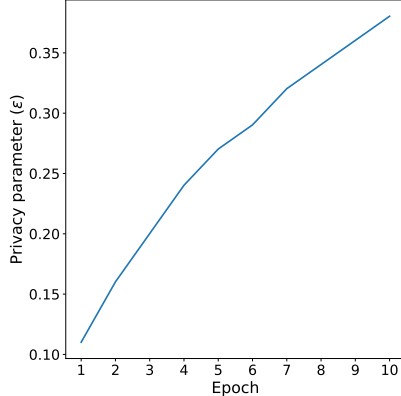

Figure 1: GeoClip results for the synthetic Gaussian dataset with 10 features. The left plot shows the average test MSE for each method over 10 epochs, with shaded regions representing the standard deviation across 20 random seeds. We observe that GeoClip achieves the fastest convergence and lowest average test MSE. The right plot shows the overall privacy budget $\varepsilon$ expended for $\delta = 10^{-5}$. This plot applies to all four algorithms, as they are tuned to achieve the same privacy level for a given number of epochs.

## 4 Experimental Results

We present empirical results demonstrating the benefits of our proposed GeoClip framework compared to AdaClip, quantile-based clipping, and standard DP-SGD. For quantile-based clipping, we use the median of per-sample norms, which has been shown to perform well across various learning tasks [10]. All $(\varepsilon, \delta)$-DP guarantees are computed using the Connect-the-Dots accountant [15]. All experiments were conducted on Google Colab using CPU resources. The code implementation is available on GitHub [16].

We start with a synthetic dataset to demonstrate how GeoClip accelerates convergence, reflecting its design motivation from Theorem 1, and then evaluate its performance on real-world datasets. All datasets are split into 80-10-10 train-validation-test sets for consistent evaluation. The results in Sections 4.1, 4.2, and 4.3 are derived from Algorithm 1. We present the results using the low-rank PCA variant in Section 4.4.

Our results indicate that our framework in Algorithm 1 is robust to the choice of hyperparameters $\beta_1$, $\beta_2$, and $h_1$. Standard values commonly used in optimization, such as $\beta_1 = 0.99$ and $\beta_2 = 0.999$, work well in our setting. The parameter $h_1$ only needs to be a small positive constant (e.g., $10^{-15}$) to ensure numerical stability. Since the eigenvalues are clamped to the range $[h_1, h_2]$ (line 13 in Algorithm 1), the trace term $\sum_i \sqrt{\lambda_i}$ is bounded between $d\sqrt{h_1}$ and $d\sqrt{h_2}$, where $d$ is the dimensionality. This allows us to set the parameter $\gamma$ to 1, as its effect can be absorbed by tuning $h_2$. For $h_2$, we have observed that the values 1 and 10 perform consistently well across datasets. Throughout all experiments, we tune only $h_2$ to select between these two options. A similar setup is used in Algorithm 2 with $\beta_2$ replaced by $\beta_3$ (set to 0.99), which is also robust across experiments.

### 4.1 Synthetic Dataset

GeoClip is designed to improve convergence, particularly in the presence of feature correlation. To empirically demonstrate this, we evaluate it on a synthetic Gaussian dataset with 20,000 samples and 10 features—five of which are correlated, while the remaining five are independent. To obtain the 5 correlated features, we first generate an 20,000 $\times$ 5 matrix $Z$ and a $5 \times 5$ matrix $A$, each with entries drawn independently from the standard normal distribution. The correlated features are obtained by multiplying $Z$ by $A$. The remaining 5 features are drawn independently from a standard multivariate normal distribution. The full feature matrix $X$ is constructed by concatenating these two blocks. The target $y$ is generated using a linear function with Gaussian noise: $y = Xw + b + \epsilon$, where $w \sim \mathcal{N}(0, I_{10})$ is a weight vector, $b \sim \mathcal{N}(0, 1)$ is a scalar bias term, and $\epsilon \sim \mathcal{N}(0, 0.01^2)$ is i.i.d. noise. We train a linear regression model using various private training methods for 10 epochs with a

batch size of 1024, tuning the learning rate for each method to ensure stable convergence. As shown in Figure 1 (left), GeoClip converges as early as epoch 2, while the next best method—quantile-based clipping—requires nearly twice as many epochs. Figure 1 (right) plots the privacy cost ($\varepsilon$) versus epoch, showing how faster convergence helps minimize overall privacy cost.

Although GeoClip introduces some computational and memory overhead, this cost is well justified by its empirical gains. GeoClip converges considerably faster than baseline methods, requiring fewer training iterations to reach comparable utility. While each iteration is slightly more expensive than in other baselines, this cost is partially offset by the reduced number of iterations. Combined with improved privacy, these benefits make the additional overhead a worthwhile trade-off.

As illustrated by the standard deviation bands in Figure 1, GeoClip not only achieves faster convergence but also demonstrates more stable training with reduced variance—an important property when privacy constraints limit training to a single run. In such scenarios, lower variability across runs enhances the reliability of the final model without requiring additional privacy budget.

## 4.2 Tabular Datasets

In addition to the synthetic dataset, we run experiments to compare the performance of different clipping strategies on three real-world datasets: Diabetes [17], Breast Cancer [18], and Android Malware [19]. We briefly describe each dataset and its corresponding learning task below.

The Diabetes dataset contains 442 samples and 10 standardized features. The continuous valued target variable indicates disease progression, making this a regression task. The Breast Cancer dataset contains 569 samples and 30 numerical features, which are standardized to zero mean and unit variance before training. The target is to predict a binary label indicating the presence or absence of cancer. The Android Malware dataset contains 4465 samples and 241 integer attributes. The target is to classify whether a program is malware or not.

For each of the three datasets, we perform a grid search over the relevant hyperparameters to identify the best model under a given $(\varepsilon, \delta)$ privacy budget. We train for 5 epochs using 20 random seeds and report the average performance along with the standard deviation in Tables 1, 2, and 3. We observe that our proposed GeoClip framework consistently outperforms all baseline methods across both regression and classification tasks. GeoClip achieves better performance with noticeably smaller standard deviations, indicating greater stability across random seeds.

Table 1: Diabetes dataset test MSE comparison for $\delta = 10^{-5}$, batch size $= 32$, and model dimension $d = 11$.

| Framework | $\varepsilon = 0.50$ | $\varepsilon = 0.86$ | $\varepsilon = 0.93$ |
|---|---|---|---|
| GeoClip (ours) | **0.073$\pm$ 0.015** | **0.044$\pm$0.003** | **0.039$\pm$0.009** |
| AdaClip | 0.077$\pm$0.027 | 0.062$\pm$0.028 | 0.055$\pm$0.014 |
| Quantile | 0.090$\pm$0.027 | 0.083$\pm$0.044 | 0.072$\pm$0.014 |
| DP-SGD | 0.108$\pm$0.040 | 0.095$\pm$0.047 | 0.072$\pm$0.040 |

Table 2: Breast Cancer dataset test accuracy (%) comparison for $\delta = 10^{-5}$, batch size $= 64$, and model dimension $d = 62$.

| Framework | $\varepsilon = 0.67$ | $\varepsilon = 0.8$ | $\varepsilon = 0.87$ |
|---|---|---|---|
| GeoClip (ours) | **87.87$\pm$3.32** | **88.57$\pm$3.37** | **93.63$\pm$1.63** |
| AdaClip | 84.90$\pm$5.91 | 85.42$\pm$5.34 | 87.71$\pm$5.87 |
| Quantile | 81.41$\pm$12.53 | 81.63$\pm$10.71 | 92.28$\pm$2.68 |
| DP-SGD | 77.32$\pm$6.17 | 79.42$\pm$9.71 | 85.95$\pm$4.45 |

Table 3: Malware dataset test accuracy (%) comparison for $\delta = 10^{-5}$, batch size $= 512$, and model dimension $d = 484$.

| Framework | $\varepsilon = 0.26$ | $\varepsilon = 0.49$ | $\varepsilon = 0.67$ |
|---|---|---|---|
| GeoClip (ours) | **90.77±1.83** | **91.64±1.26** | **92.67±1.63** |
| AdaClip | 88.35±3.27 | 90.25±1.33 | 90.23±3.11 |
| Quantile | 77.84±1.29 | 78.84±1.27 | 81.86±1.31 |
| DP-SGD | 88.04±2.21 | 90.55±1.55 | 90.57±1.61 |

## 4.3 Final Layer Fine-Tuning

In many transfer learning scenarios, fine-tuning only the final layer is standard practice due to both its computational efficiency and minimal privacy cost. Last-layer fine-tuning is a well-suited application of GeoClip, as it involves a small number of trainable parameters, making covariance estimation more tractable. This setup also benefits from GeoClip's faster convergence, which is particularly valuable in privacy-constrained settings where only limited training iterations are feasible.

To demonstrate this, we design an experiment where a convolutional neural network (CNN) is first trained on MNIST [20] using the Adam optimizer and then transferred to Fashion-MNIST [21] by freezing all layers except the final fully connected layer. The CNN consists of two convolutional and pooling layers followed by a linear compression layer that reduces the feature size to 50, resulting in a total of only 510 trainable parameters. We fine-tune this layer using different methods under varying privacy budgets, and present the results in Table 4.

Table 4: Final layer DP fine-tuning on Fashion-MNIST for 4 epochs and $\delta = 10^{-6}$ over 5 seeds.

| Framework | $\varepsilon = 0.6$ | $\varepsilon = 1$ |
|---|---|---|
| GeoClip (Ours) | **73.09±0.72** | **73.09±0.63** |
| AdaClip | 68.35±0.41 | 69.24±0.28 |
| Quantile-based | 71.78±1.28 | 72.09±1.12 |
| DP-SGD | 69.40±0.82 | 69.83±0.83 |

## 4.4 Low-Rank PCA Results

To evaluate our low-rank PCA algorithm (Algorithm 2), we construct a synthetic binary classification dataset with 20,000 samples and 400 Gaussian features, where 50 are correlated and 350 are uncorrelated. The feature matrix is constructed in the same manner as in Section 4.1. Labels are generated by applying a linear function to the features, adding Gaussian noise, and thresholding the sigmoid output. We train a logistic regression model with 802 parameters and compare our method against existing benchmarks using a rank-50 low-rank PCA approximation. As shown in the left panel of Figure 2, our approach converges faster than competing methods, even with this low-rank approximation.

We also evaluate our method on the USPS dataset [22] using logistic regression with 2,570 trainable parameters (256 input features $\times$ 10 classes + 10 biases). The USPS dataset contains 9,298 grayscale handwritten digit images (0–9), each of size $16 \times 16$ pixels. This compact benchmark is commonly used for evaluating digit classification models. For this dataset, we apply a low-rank PCA approximation with rank 100. Results are shown in the right panel of Figure 2. As with the synthetic dataset, GeoClip with low-rank PCA also achieves faster convergence on USPS compared to baseline methods. We provide $\varepsilon$-vs-iteration plots for both datasets, illustrating how faster convergence reduces overall privacy cost in Appendix D.

We compare the accuracy of GeoClip using Algorithm 1 and Algorithm 2 with $k = 50$ on the same synthetic dataset, with the results shown in Figure 3 in Appendix D. The results demonstrate that even with $k = 50$, Algorithm 2 achieves accuracy comparable to Algorithm 1, which computes the full covariance.

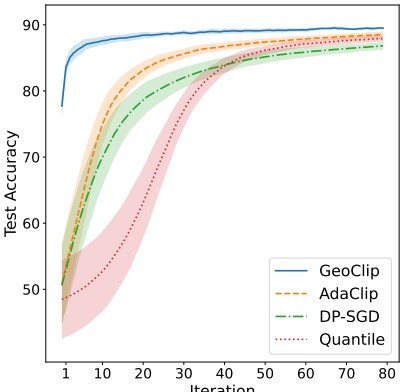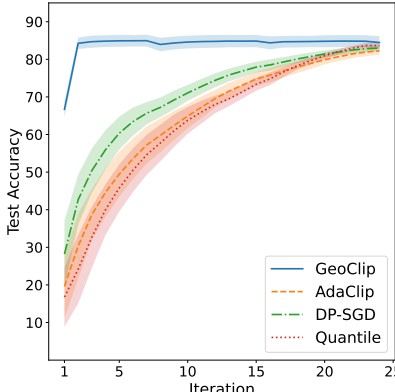

Figure 2: The left panel shows results on the synthetic Gaussian dataset with 400 features using a rank-50 PCA approximation for GeoClip. The plot displays average test accuracy $(\%)$ over 80 iterations with a batch size of 1024. GeoClip achieves the fastest convergence and highest average accuracy. The right panel shows results on the USPS dataset using a rank-100 approximation over 25 iterations with a batch size of 1024, where a similar convergence trend is observed. Shaded regions represent standard deviation across 20 random seeds.

We also conduct an ablation study on $\gamma$, $h_1$, and $h_2$ using the USPS dataset under the same experimental settings. The results are presented in Appendix E. The findings show that varying $\gamma$ results in only minor performance changes. Likewise, $h_1$ has no noticeable impact, and $h_2$ introduces small variations.

## 5 Conclusions

We have introduced GeoClip, a geometry-aware framework for differentially private SGD that leverages the structure of the gradient distribution to improve both utility and convergence. By operating in a basis adapted to the estimated noisy gradients, GeoClip injects noise more strategically, thereby reducing distortion without incurring additional privacy cost. We have provided a formal analysis of convergence guarantees which characterizes the optimal transformation. Our empirical results on synthetic and real-world datasets show that GeoClip consistently converges faster and outperforms existing adaptive clipping methods, improving both the mean and standard deviation of the performance metrics over multiple runs. Via low-rank approximation method, we have shown that GeoClip scales to the high-dimensional data setting, thus making it suitable for practical deployment in large, privacy-sensitive models.

**Limitations.** The linear transformation involved in GeoClip requires an additional computation via an eigendecomposition. Our low-rank approximation addresses that to some extent. The algorithm introduces additional hyperparameters (most notably $h_2$) compared to standard DP-SGD, which must be tuned for optimal performance. Our experiments have been performed on a limited collection of datasets; additional testing is needed to see how our algorithm performs in more generality.

**Broader impact.** As discussed in the Introduction, our work is motivated by societal concerns, with a focus on improving the theoretical limits of differentially private optimization.

## Acknowledgements

We thank the anonymous reviewers for their valuable feedback, which significantly improved the quality of this paper. This work was supported by the National Science Foundation under Grant Nos. CIF-1901243, CIF-2312666, and CIF-2007688.

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

# A  Proof of Theorem 1

**Theorem** (Convergence of GeoClip). *Assume $f$ has an $L$-Lipschitz continuous gradient. Further, assume the stochastic gradients are bounded, i.e., $\|\nabla f_k(\theta)\| \leq G$, and have bounded variance, i.e., $\mathbb{E}_k\|\nabla f_k(\theta) - \nabla f(\theta)\|^2 \leq \sigma_g^2$. Let $\theta^* = \arg\min_{\theta \in \mathbb{R}^d} f(\theta)$ denote the optimal solution, and suppose the learning rate satisfies $\eta < \frac{2}{3L}$. Then, for the iterates $\{\theta_t\}_{t=0}^{T-1}$ produced by GeoClip with batch size 1 using the update rule $\theta_{t+1} = \theta_t - \eta\tilde{g}_t$, where $\tilde{g}_t$ is defined in (5), the average squared gradient norm satisfies:*

$$\frac{1}{T}\sum_{t=0}^{T-1}\mathbb{E}\|\nabla f(\theta_t)\|^2 \leq \underbrace{\frac{f(\theta_0) - f(\theta^*)}{T\left(\eta - \frac{3L\eta^2}{2}\right)}}_{\text{Optimization gap}} + \underbrace{\frac{3L\eta}{2 - 3L\eta}\sigma_g^2}_{\text{Gradient variance term}} + \underbrace{\frac{L\eta\sigma^2}{T(2-3L\eta)}\sum_{t=0}^{T-1}\mathbb{E}\,\text{Tr}\left[\left(M_t^\top M_t\right)^{-1}\right]}_{\text{Noise-injection term}}$$

$$+ \underbrace{\frac{2}{T(2-3L\eta)}\sum_{t=0}^{T-1}\mathbb{E}\left[\beta(a_t)\left(\text{Tr}\left(M_t^\top M_t \Sigma_t\right) + \|M_t(\mathbb{E}[g_t \mid \theta^t] - a_t)\|^2\right)\right]}_{\text{Clipping error term}}, \tag{23}$$

*where $\theta^t = (\theta_0, \ldots, \theta_t)$ represents the history of parameter values up to iteration $t$, $\Sigma_t = \text{Cov}(g_t|\theta^t)$, and*

$$\beta(a_t) = (G + \|a_t\|)\left(G + \frac{3L\eta}{2}(G + \|a_t\|)\right). \tag{24}$$

*Proof.* We can express the noisy gradient as:

$$\tilde{g}_t = M_t^{-1}\tilde{\omega}_t + a_t \tag{25}$$

$$= \frac{M_t^{-1}\omega_t}{\max\{1, \|\omega_t\|\}} + M_t^{-1}N_t + a_t \tag{26}$$

$$= \frac{g_t - a_t}{\max\{\|M_t(g_t - a_t)\|, 1\}} + M_t^{-1}N_t + a_t. \tag{27}$$

Thus, the parameter update takes the form

$$\theta_{t+1} = \theta_t - \eta\tilde{g}_t \tag{28}$$

$$= \theta_t - \eta\left[\frac{g_t - a_t}{\max\{\|M_t(g_t - a_t)\|, 1\}} + M_t^{-1}N_t + a_t\right]. \tag{29}$$

We define the following quantities:

$$c_t = \frac{g_t - a_t}{\max\{\|M_t(g_t - a_t)\|, 1\}}, \tag{30}$$

$$\Delta_t = c_t - (g_t - a_t). \tag{31}$$

$c_t$ is the clipped version of $g_t - a_t$. $\Delta_t$ quantifies the distortion due to clipping, being zero when no clipping occurs and negative when clipping is applied. By the $L$-Lipschitz continuity of the gradient of $f(\theta)$, we have:

$$f(\theta_{t+1}) \leq f(\theta_t) + \langle\nabla f(\theta_t), \theta_{t+1} - \theta_t\rangle + \frac{L}{2}\|\theta_{t+1} - \theta_t\|^2. \tag{32}$$

Let $\theta^t = (\theta_0, \ldots, \theta_t)$ describe the entire history of states. Taking the expectation conditioned on $\theta^t$, we obtain:

$$\mathbb{E}\left[f\left(\theta_{t+1}\right)\middle|\theta^t\right]$$

$$\leq f\left(\theta_t\right) + \mathbb{E}\left[\langle\nabla f\left(\theta_t\right), \theta_{t+1} - \theta_t\rangle\middle|\theta^t\right] + \frac{L}{2}\mathbb{E}\left[\|\theta_{t+1} - \theta_t\|^2\middle|\theta^t\right] \tag{33}$$

$$= f\left(\theta_t\right) - \eta\mathbb{E}\left[\langle\nabla f\left(\theta_t\right), c_t + M_t^{-1}N_t + a_t\rangle\middle|\theta^t\right] + \frac{L\eta^2}{2}\mathbb{E}\left[\|c_t + M_t^{-1}N_t + a_t\|^2\middle|\theta^t\right] \tag{34}$$

$$= f\left(\theta_t\right) - \eta\mathbb{E}\left[\langle\nabla f\left(\theta_t\right), c_t + a_t\rangle\middle|\theta^t\right] + \frac{L\eta^2}{2}\mathbb{E}\left[\|c_t + a_t\|^2\middle|\theta^t\right] + \frac{L\eta^2}{2}\mathbb{E}\left[\|M_t^{-1}N_t\|^2\middle|\theta^t\right] \tag{35}$$

$$= f\left(\theta_t\right) - \eta\mathbb{E}\left[\langle\nabla f\left(\theta_t\right), c_t + a_t\rangle\middle|\theta^t\right] + \frac{L\eta^2}{2}\mathbb{E}\left[\|c_t + a_t\|^2\middle|\theta^t\right] + \frac{L\eta^2\sigma^2}{2}\left\|M_t^{-1}\right\|_F^2 \tag{36}$$

$$= f\left(\theta_t\right) - \eta\mathbb{E}\left[\langle\nabla f\left(\theta_t\right), g_t + \Delta_t\rangle\middle|\theta^t\right] + \frac{L\eta^2}{2}\mathbb{E}\left[\|g_t + \Delta_t\|^2\middle|\theta^t\right] + \frac{L\eta^2\sigma^2}{2}\left\|M_t^{-1}\right\|_F^2 \tag{37}$$

$$= f\left(\theta_t\right) - \eta\|\nabla f\left(\theta_t\right)\|^2 - \eta\mathbb{E}\left[\langle\nabla f\left(\theta_t\right), \Delta_t\rangle\middle|\theta^t\right]$$
$$+ \frac{L\eta^2}{2}\mathbb{E}\left[\|g_t - \nabla f\left(\theta_t\right) + \nabla f\left(\theta_t\right) + \Delta_t\|^2\middle|\theta^t\right] + \frac{L\eta^2\sigma^2}{2}\left\|M_t^{-1}\right\|_F^2 \tag{38}$$

where (35) follows from $\mathbb{E}[N_t] = 0$ and the independence of $N_t$ from $g_t$ (and thus from $c_t + a_t$). The equality (36) follows because $\mathbb{E}[NN^\top] = \sigma^2 I_d$, and $\|M_t^{-1}\|_F$ represents the Frobenius norm of $M_t^{-1}$. The equality (37) follows from $c_t + a_t = \Delta_t + g_t$. The last equality follows because

$$\mathbb{E}\left[\langle\nabla f\left(\theta_t\right), g_t + \Delta_t\rangle\middle|\theta^t\right] = \mathbb{E}\left[\langle\nabla f\left(\theta_t\right), \Delta_t\rangle\middle|\theta^t\right] + \mathbb{E}\left[\langle\nabla f\left(\theta_t\right), g_t\rangle\middle|\theta^t\right] \tag{39}$$

$$= \mathbb{E}\left[\langle\nabla f\left(\theta_t\right), \Delta_t\rangle\middle|\theta^t\right] + \nabla f\left(\theta_t\right)^\top\mathbb{E}[g_t|\theta^t] \tag{40}$$

$$= \mathbb{E}\left[\langle\nabla f\left(\theta_t\right), \Delta_t\rangle\middle|\theta^t\right] + \|\nabla f\left(\theta_t\right)\|^2, \tag{41}$$

where (41) follows because $g_t$ is an unbiased estimator of the true gradient, i.e., $\mathbb{E}[g_t \mid \theta^t] = \nabla f(\theta_t)$. From Jensen's inequality, we have:

$$\mathbb{E}\left[\|g_t - \nabla f(\theta_t) + \nabla f(\theta_t) + \Delta_t\|^2\middle|\theta^t\right]$$
$$\leq 3\left(\mathbb{E}\left[\|g_t - \nabla f(\theta_t)\|^2\middle|\theta^t\right] + \|\nabla f(\theta_t)\|^2 + \mathbb{E}\left[\|\Delta_t\|^2\middle|\theta^t\right]\right). \tag{42}$$

From the Cauchy–Schwarz inequality, we have:

$$\mathbb{E}\left[\langle\nabla f(\theta_t), \Delta_t\rangle\middle|\theta^t\right] \leq \|\nabla f(\theta_t)\|\,\mathbb{E}\left[\|\Delta_t\|\middle|\theta^t\right]. \tag{43}$$

Plugging (42) and (43) into (38), we obtain:

$$\mathbb{E}\left[f\left(\theta_{t+1}\right)\middle|\theta^t\right] \leq f\left(\theta_t\right) - \eta\|\nabla f\left(\theta_t\right)\|^2 + \eta\|\nabla f\left(\theta_t\right)\|\,\mathbb{E}\left[\|\Delta_t\|\middle|\theta^t\right]$$
$$+ \frac{3L\eta^2}{2}\left[\mathbb{E}\left[\|g_t - \nabla f(\theta_t)\|^2\middle|\theta^t\right] + \|\nabla f(\theta_t)\|^2 + \mathbb{E}\left[\|\Delta_t\|^2\middle|\theta^t\right]\right] + \frac{L\eta^2\sigma^2}{2}\left\|M_t^{-1}\right\|_F^2. \tag{44}$$

To bound $\mathbb{E}\left[\|\Delta_t\|\middle|\theta^t\right]$ and $\mathbb{E}\left[\|\Delta_t\|^2\middle|\theta^t\right]$, we first bound $\Pr\left(\|\Delta_t\| > 0\middle|\theta^t\right)$ and $\|\Delta_t\|$ given $\theta^t$ and $\|\Delta_t\| > 0$. Using Markov's inequality, we obtain:

$$\Pr\left(\|\Delta_t\| > 0\middle|\theta^t\right) = \Pr\left(\|M_t(g_t - a_t)\|^2 > 1\middle|\theta^t\right) \tag{45}$$

$$\leq \Pr\left(\|M_t(g_t - a_t)\|^2 \geq 1\middle|\theta^t\right) \tag{46}$$

$$\leq \mathbb{E}\left[\|M_t(g_t - a_t)\|^2\middle|\theta^t\right] \tag{47}$$

$$= \mathbb{E}\left[\|M_t\left(g_t - \mathbb{E}[g_t|\theta^t] + \mathbb{E}[g_t|\theta^t] - a_t\right)\|^2\middle|\theta^t\right] \tag{48}$$

$$= \mathbb{E}\left[\|M_t\left(g_t - \mathbb{E}[g_t|\theta^t]\right)\|^2\middle|\theta^t\right] + \|M_t(\mathbb{E}[g_t|\theta^t] - a_t)\|^2 \tag{49}$$

$$= \text{Tr}\left(\text{Cov}\left(M_t g_t\middle|\theta^t\right)\right) + \|M_t(\mathbb{E}[g_t|\theta^t] - a_t)\|^2. \tag{50}$$

Using (30) and (31), we obtain:

$$\|\Delta_t\| = \left\|\left(\frac{1}{\max\{\|M_t(g_t - a_t)\|, 1\}} - 1\right)(g_t - a_t)\right\| \tag{51}$$

$$\leq \|g_t - a_t\| \tag{52}$$

$$\leq \|g_t\| + \|a_t\| \tag{53}$$

$$\leq G + \|a_t\|, \tag{54}$$

where the final inequality follows from $\|\nabla f_k(\theta)\| \leq G$. Therefore,

$$\mathbb{E}\left[\|\Delta_t\|\,\middle|\,\theta^t\right] = \Pr\left(\|\Delta_t\| > 0\,\middle|\,\theta^t\right)\mathbb{E}\left[\|\Delta_t\|\,\middle|\,\|\Delta_t\| > 0, \theta^t\right] \tag{55}$$

$$\leq (G + \|a_t\|)\left(\mathrm{Tr}\left(\mathrm{Cov}\left(M_t g_t\,\middle|\,\theta^t\right)\right) + \|M_t(\mathbb{E}[g_t\,|\,\theta^t] - a_t)\|^2\right). \tag{56}$$

Similarly, we obtain the following bound:

$$\mathbb{E}\left[\|\Delta_t\|^2\,\middle|\,\theta^t\right] \leq (G + \|a\|)^2\left(\mathrm{Tr}\left(\mathrm{Cov}\left(M_t g_t\,\middle|\,\theta^t\right)\right) + \|M_t(\mathbb{E}[g_t\,|\,\theta^t] - a_t)\|^2\right). \tag{57}$$

By plugging (56) and (57) into (44) and rearranging the terms, we obtain

$\mathbb{E}\left[f\left(\theta_{t+1}\right)\middle|\theta^t\right]$

$$\leq f\left(\theta_t\right) + \left(\frac{3L\eta^2}{2} - \eta\right)\|\nabla f(\theta_t)\|^2 + \eta(G + \|a_t\|)\left(G + \frac{3L\eta}{2}(G + \|a_t\|)\right)$$

$$\left[\mathrm{Tr}\left(\mathrm{Cov}\left(M_t g_t\,\middle|\,\theta^t\right)\right) + \|M_t(\mathbb{E}[g_t\,|\,\theta^t] - a_t)\|^2\right] + \frac{3L\eta^2}{2}\mathbb{E}\left[\|g_t - \nabla f(\theta_t)\|^2\,\middle|\,\theta^t\right] + \frac{L\eta^2\sigma^2}{2}\|M_t^{-1}\|_F^2. \tag{58}$$

Above, we have used the fact that $\|\nabla f(\theta_t)\| \leq G$, which itself follows from Jensen's inequality and the assumption that $\|\nabla f_k(\theta)\| \leq G$. Let

$$\beta(a_t) = (G + \|a_t\|)\left(G + \frac{3L\eta}{2}(G + \|a_t\|)\right). \tag{59}$$

Rearranging the terms, applying the law of total expectation (now taking expectation over the entire history of states $\theta^t$), and using the bound $\mathbb{E}_k\|\nabla f_k(\theta) - \nabla f(\theta)\|^2 \leq \sigma_g^2$, we obtain:

$$\left(\eta - \frac{3L\eta^2}{2}\right)\mathbb{E}\|\nabla f(\theta_t)\|^2 \leq \mathbb{E}f(\theta_t) - \mathbb{E}f(\theta_{t+1}) + \frac{3L\eta^2\sigma_g^2}{2} + \frac{L\eta^2\sigma^2}{2}\mathbb{E}\|M_t^{-1}\|_F^2$$

$$+ \eta\,\mathbb{E}\left[\beta(a_t)\left(\mathrm{Tr}\left(\mathrm{Cov}\left(M_t g_t\,\middle|\,\theta^t\right)\right) + \|M_t(\mathbb{E}[g_t\,|\,\theta^t] - a_t)\|^2\right)\right]. \tag{60}$$

Summing over $t = 0$ to $T - 1$ and applying the telescoping sum, we get:

$$\left(\eta - \frac{3L\eta^2}{2}\right)\sum_{t=0}^{T-1}\mathbb{E}\|\nabla f(\theta_t)\|^2 \leq f(\theta_0) - \mathbb{E}f(\theta_T) + \frac{3L\eta^2\sigma_g^2}{2}T + \frac{L\eta^2\sigma^2}{2}\sum_{t=0}^{T-1}\mathbb{E}\|M_t^{-1}\|_F^2$$

$$+ \eta\sum_{t=0}^{T-1}\mathbb{E}\left[\beta(a_t)\left(\mathrm{Tr}\left(\mathrm{Cov}\left(M_t g_t\,\middle|\,\theta^t\right)\right) + \|M_t(\mathbb{E}[g_t\,|\,\theta^t] - a_t)\|^2\right)\right] \tag{61}$$

Dividing by $T$, assuming $\eta \leq \frac{2}{3L}$, applying the identity $\|M_t^{-1}\|_F^2 = \mathrm{Tr}\left(\left(M_t^\top M_t\right)^{-1}\right)$, and using the fact that $\mathbb{E}f(\theta_T) \geq f(\theta^*)$, we obtain:

$$\frac{1}{T}\sum_{t=0}^{T-1}\mathbb{E}\|\nabla f(\theta_t)\|^2$$

$$\leq \frac{f(\theta_0) - f(\theta^*)}{T\left(\eta - \frac{3L\eta^2}{2}\right)} + \frac{3L\eta}{2 - 3L\eta}\sigma_g^2 + \frac{L\eta\sigma^2}{T(2 - 3L\eta)}\sum_{t=0}^{T-1}\mathbb{E}\left[\mathrm{Tr}\left(M_t^\top M_t\right)^{-1}\right]$$

$$+ \frac{2}{T(2 - 3L\eta)}\sum_{t=0}^{T-1}\mathbb{E}\left[\beta(a_t)\left(\mathrm{Tr}\left(M_t^\top M_t \mathrm{Cov}(g_t\,|\,\theta^t)\right) + \|M_t(\mathbb{E}[g_t\,|\,\theta^t] - a_t)\|^2\right)\right]. \tag{62}$$

$\square$

# B Proof of Theorem 2

We would like to solve the following optimization

$$\underset{M_t}{\text{minimize}} \quad \text{Tr}\left(M_t^\top M_t\right)^{-1}$$

$$\text{subject to} \quad \text{Tr}\left(M_t^\top M_t \Sigma_t\right) \leq \gamma. \tag{63}$$

Defining $A_t$ as the Gram matrix of $M_t$, i.e., $A_t = M_t^\top M_t$, we can reformulate the optimization problem as:

$$\underset{A_t}{\text{minimize}} \quad \text{Tr}(A_t^{-1})$$

$$\text{subject to} \quad \text{Tr}(A_t \Sigma_t) \leq \gamma. \tag{64}$$

Both the objective and the constraint are convex in $A$, so to solve the problem, we introduce the Lagrangian function:

$$\ell(A_t, \mu) = \text{Tr}(A_t^{-1}) + \mu\left(\text{Tr}(A_t \Sigma_t) - \gamma\right), \tag{65}$$

Taking the derivative of $\ell(A_t, \mu)$ with respect to $A_t$ and setting the derivative to zero for optimality, we get

$$A_t^{-2} = \mu \, \Sigma_t. \tag{66}$$

and so

$$A_t = \frac{1}{\sqrt{\mu}} \Sigma_t^{-\frac{1}{2}}. \tag{67}$$

Substituting $A_t$ in the constraint $\text{Tr}\left(A_t \Sigma_t\right) = \gamma$, we get

$$\frac{1}{\sqrt{\mu}} \text{Tr}\left(\Sigma_t^{\frac{1}{2}}\right) = \gamma. \tag{68}$$

Solving for $\mu$:

$$\sqrt{\mu} = \frac{1}{\gamma} \text{Tr}\left(\Sigma_t^{\frac{1}{2}}\right). \tag{69}$$

Thus, the optimal $A_t$ is:

$$A_t = \frac{\gamma}{\text{Tr}\left(\Sigma_t^{\frac{1}{2}}\right)} \Sigma_t^{-\frac{1}{2}}. \tag{70}$$

Using the eigen decomposition, we write $\Sigma_t$ as:

$$\Sigma_t = U_t \Lambda_t U_t^\top, \tag{71}$$

where $U_t$ is an orthogonal matrix whose columns are the eigenvectors of $\Sigma_t$, and

$$\Lambda_t = \text{diag}(\lambda_1, \ldots, \lambda_d)$$

is a diagonal matrix containing the corresponding eigenvalues. We now have

$$\Sigma_t^{\frac{1}{2}} = U_t \Lambda_t^{\frac{1}{2}} U_t^\top, \quad \Sigma_t^{-\frac{1}{2}} = U_t \Lambda_t^{-\frac{1}{2}} U_t^\top, \tag{72}$$

Thus, the final expression for $A_t$ is:

$$A_t = \frac{\gamma}{\text{Tr}(\Lambda_t^{\frac{1}{2}})} U_t \Lambda_t^{-\frac{1}{2}} U_t^\top. \tag{73}$$

Therefore, we have

$$M_t^\top M_t = \frac{\gamma}{\text{Tr}(\Lambda_t^{\frac{1}{2}})} U_t \Lambda_t^{-\frac{1}{2}} U_t^\top, \tag{74}$$

and

$$M_t = \left(\frac{\gamma}{\text{Tr}(\Lambda_t^{\frac{1}{2}})}\right)^{1/2} \Lambda_t^{-\frac{1}{4}} U_t^\top. \tag{75}$$

Since $\text{Tr}(\Lambda_t^{1/2}) = \sum_{i=1}^d \sqrt{\lambda_i}$, we can simplify $M_t$ as follows:

$$M_t = \left(\frac{\gamma}{\sum_{i=1}^d \sqrt{\lambda_i}}\right)^{1/2} \Lambda_t^{-1/4} U_t^\top. \tag{76}$$

**Algorithm 3** GEOCLIP WITH RANK-$k$ PCA

---

**Require:** Dataset $\mathcal{D}$, model $f_\theta$, loss $\mathcal{L}$, learning rate $\eta$, noise scale $\sigma$, steps $T$, rank $k$, hyperparameters $h_1, h_2, \beta_1, \beta_3$

1: Initialize $\theta$, mean vector $a_0 = 0$, $U_0 = [e_1, \ldots, e_k]$ where $e_i \in \mathbb{R}^d$ is the $i$-th standard basis vector, $\Lambda_0 = I_k$
2: Compute transform $M_0 \leftarrow (\gamma/k)^{1/2}\, \Lambda_0^{-1/4}U_0^\top$ and $M_0^{\text{inv}} \leftarrow (\gamma/k)^{-1/2}\, U_0\Lambda_0^{1/4}$
3: **for** $t = 0$ to $T$ **do**
4:      Sample a data point $(x_t, y_t)$
5:      Compute gradient $g_t \leftarrow \nabla_\theta \mathcal{L}(f_\theta(x_t), y_t)$
6:      Center and transform: $\omega_t \leftarrow M_t(g_t - a_t)$
7:      Clip: $\bar{\omega}_t \leftarrow \omega_t / \max(1, \|\omega_t\|_2)$
8:      Add noise: $\tilde{\omega}_t \leftarrow \bar{\omega}_t + N$, where $N \sim \mathcal{N}(0, \sigma^2 I_k)$
9:      Map back: $\tilde{g}_t \leftarrow M_t^{\text{inv}}\tilde{\omega}_t + a_t$
10:     Update model: $\theta_{t+1} \leftarrow \theta_t - \eta\tilde{g}_t$
11:     Update mean: $a_{t+1} \leftarrow \beta_1 a_t + (1 - \beta_1)\tilde{g}_t$
12:     Update eigenspace: $(U_{t+1}, \Lambda_{t+1}) \leftarrow$ STREAMING RANK-$k$ PCA$(U_t, \Lambda_t, \tilde{g}_t, a_{t+1}, \beta_3, k)$
13:     Clamp eigenvalues: $\lambda_i \leftarrow \texttt{Clamp}(\lambda_i, \min = h_1, \max = h_2)$
14:     Set $M_{t+1} \leftarrow \left(\gamma/\sum_i \sqrt{\lambda_i}\right)^{1/2} \Lambda_{t+1}^{-1/4}U_{t+1}^\top$
15:     Set $M_{t+1}^{\text{inv}} \leftarrow \left(\gamma/\sum_i \sqrt{\lambda_i}\right)^{-1/2} U_{t+1}\Lambda_{t+1}^{1/4}$
16: **end for**
17: **return** Final parameters $\theta$

---

## C   GeoClip with Low-Rank PCA

Computing and storing the full gradient covariance matrix becomes infeasible in high dimensions. To address this, we propose a low-rank approximation method in Algorithm 3, which incorporates a rank-$k$ PCA step described in the STREAMING RANK-$k$ PCA algorithm.

To compute $M_0$ and $M_0^{\text{inv}}$ in line 2 of Algorithm 3, we use the simplification $\text{Tr}(\Lambda_0^{1/2}) = \sum_{i=1}^k 1 = k$. Also, in line 13, each $\lambda_i$ depends on $t$, but we omit the subscript for notational simplicity.

---

STREAMING RANK-$k$ PCA

---

**Require:** Eigenvectors $U_t \in \mathbb{R}^{d \times k}$, eigenvalues $\Lambda_t \in \mathbb{R}^{k \times k}$, gradient $\tilde{g}_t \in \mathbb{R}^d$, mean $a_{t+1} \in \mathbb{R}^d$, factor $\beta_3 \in \mathbb{R}$, rank $k$

1: Center: $z \leftarrow \tilde{g}_t - a_{t+1}$
2: Form augmented matrix: $U_{\text{aug}} \leftarrow [U_t \;\; z]$
3: Compute: $Z \leftarrow U_{\text{aug}} \, \text{diag}(\sqrt{\beta_3\lambda_1}, \ldots, \sqrt{\beta_3\lambda_k}, \sqrt{1 - \beta_3})$
4: Perform SVD: $Z = VSR^\top$
5: Set $U_{t+1} \leftarrow$ first $k$ columns of $V$
6: Set $\Lambda_{t+1} \leftarrow$ squares of the first $k$ singular values in $S$
7: Return: $U_{t+1}, \Lambda_{t+1}$

---

## D   Additional Plots Related to Section 4.4

The middle and left panels of Figure 3 show the privacy cost ($\varepsilon$) versus iteration curves corresponding to the accuracy–iteration plots in Figure 2 (Section 4.4, Low-Rank PCA Results). These plots highlight how faster convergence reduces overall privacy cost. For example, on the USPS dataset (Figure 2), GeoClip achieves high accuracy within the first few iterations, requiring only $\varepsilon \approx 0.15$, whereas quantile-based clipping takes about 24 iterations to reach similar accuracy, incurring a higher privacy cost of $\varepsilon \approx 0.5$.

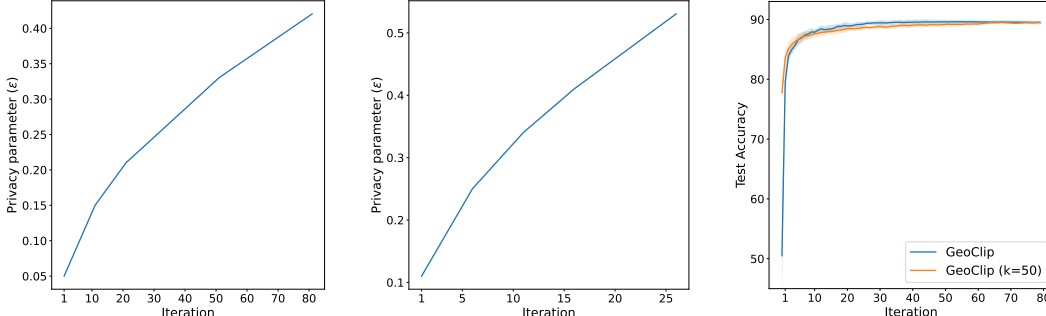

Figure 3: The left plot shows the overall privacy budget $\varepsilon$ spent on training the synthetic Gaussian dataset with 400 features for $\delta = 10^{-5}$, while the middle plot shows the same for the USPS dataset with $\delta = 10^{-5}$. These plots apply to all four algorithms, which are tuned to achieve the same privacy level for a given number of iterations. The right panel compares GeoClip's performance using the full-covariance and low-rank approximation algorithms.

We compare the accuracy of GeoClip using Algorithm 1 and Algorithm 2 with $k = 50$ on the same synthetic dataset described in Section 4.4, as shown in the right panel of Figure 3. The results show that even with $k = 50$, Algorithm 2 achieves accuracy comparable to Algorithm 1, which computes the full covariance.

# E   Ablation Study

We conduct an ablation study on $\gamma$, $h_1$, and $h_2$ using the USPS dataset, under the same settings as in Section 4.4. As shown, varying $\gamma$ does not significantly affect performance. The results indicate no noticeable impact from $h_1$, and only minor variations with $h_2$.

Table 5: Test accuracy (%) for different values of $\gamma$ on the USPS dataset with $h_2 = 10$ and $h_1 = 10^{-15}$

| Method | $\gamma = 0.2$ | $\gamma = 0.6$ | $\gamma = 1.0$ |
|---|---|---|---|
| GeoClip ($k = 50$) | 85.537 | 86.182 | 85.752 |

Table 6: Test accuracy (%) for different values of $h_2$ on the USPS dataset with $h_1 = 10^{-15}$ and $\gamma = 1$

| Method | $h_2 = 1$ | $h_2 = 10$ | $h_2 = 100$ | $h_2 = 1000$ |
|---|---|---|---|---|
| GeoClip ($k = 50$) | 86.021 | 85.752 | 85.752 | 85.752 |

Table 7: Test accuracy (%) for different values of $h_1$ on the USPS dataset with $h_2 = 10$ and $\gamma = 1$

| Method | $h_1 = 10^{-6}$ | $h_1 = 10^{-10}$ | $h_1 = 10^{-15}$ |
|---|---|---|---|
| GeoClip ($k = 50$) | 85.752 | 85.752 | 85.752 |

