# OpenReview forum: "GeoClip: Geometry-Aware Clipping for Differentially Private SGD"
_NeurIPS.cc/2025/Conference — NeurIPS 2025 poster_

### Official Review · Reviewer_ocmc · 2025-06-17

**Clarity:** 4
**Significance:** 4
**Originality:** 4
**Rating:** 5
**Confidence:** 4

**Summary:**

This paper introduces a novel method for adaptive clipping in large-scale differentially private optimization. The proposed approach accounts for correlations across gradient vectors, specifically addressing scenarios where gradients exhibit varying underlying geometries. The authors provide theoretical guarantees for their method and demonstrate improved empirical performance over existing algorithms that do not incorporate geometric information during the clipping process.

**Questions:**

1. **Convergence Under Approximation**:  What is the final convergence rate of GeoClip when combined with streaming PCA? How does the convergence analysis differ from the case without approximation? Additionally, under a low-rank Hessian assumption as explored in [1,2], is it possible to obtain improved dependencies in the utility gap, akin to the theoretical gains demonstrated in those works?

2. **Utility of Chosen Preconditioner**: In Theorem 2, what is the final utility guarantee corresponding to the specific choice of the preconditioning matrix $M_t$? Is this choice theoretically optimal in terms of convergence or privacy-utility trade-off?

3. **Faster Preconditioning**:  Is it possible to prove faster convergence to optimal utility by leveraging geometry-aware techniques? More specifically, can one characterize the optimal choice of $M_t$ that minimizes the number of iterations $T$ required to achieve a desired utility, and does this lead to any fundamental improvements in convergence rate?

---

**References**:
[1] Li, X., Liu, D., Hashimoto, T. B., Inan, H. A., Kulkarni, J., Lee, Y. T., & Guha Thakurta, A. (2022). *When does differentially private learning not suffer in high dimensions?* Advances in Neural Information Processing Systems, 35, 28616–28630.
[2] Zhang, L., Li, B., Thekumparampil, K. K., Oh, S., & He, N. (2023). *DPZero: Private fine-tuning of language models without backpropagation.* arXiv preprint arXiv:2310.09639.

**Ethical Concerns:**

["NO or VERY MINOR ethics concerns only"]

**Final Justification:**

The authors responded to the comments effectively, and I remain confident in the score I assigned. While I initially found the convergence analysis somewhat confusing, the paper overall makes a strong contribution. The authors' feedback helped clarify the individual components of the analysis and how the approach preserves the standard DP-SGD guarantees, despite the algorithm’s complexity. That said, as noted by the authors, some theoretical aspects of the work remain open for future exploration and may require more advanced technical tools to fully grasp. Nonetheless, the intuition provided is solid and the paper offers a novel perspective, particularly in its geometry-aware approach, which has not been explored before.

**Limitations:**

Yes

**Quality:**

3

**Strengths And Weaknesses:**

### **Strengths**

1. **Clarity and Theoretical Foundation**: The paper is clearly written and provides a well-structured overview of the theoretical underpinnings of the proposed algorithm. This clarity allows the reader to follow the motivation, construction, and justification of the technique with ease.

2. **Innovative Use of Statistical Tools**: The authors successfully integrate concepts from algorithmic statistics—particularly streaming PCA—into the framework of adaptive clipping. This combination leads to a novel and principled approach to differentially private optimization.

3. **Strong Empirical Performance**: The proposed method demonstrates empirical improvements over existing baselines on standard benchmark tasks, showcasing its practical effectiveness.

4. **Theoretical Depth and Ablation**: The paper includes thoughtful theoretical components such as the use of low-rank approximation, which are well-explained and demonstrate the authors' careful consideration of the algorithm's efficiency and design.

5. **Novel Impact**: The algorithm addresses a meaningful gap in the literature by handling general gradient geometries in a computationally aware manner. This increases the practicality of adaptive clipping in broader settings.

---

### **Weaknesses**

1. **Resource Overhead**: A primary limitation of the method is its computational and memory overhead. Storing and updating a full covariance matrix of gradients can be prohibitive, especially in large-scale settings such as training language models. Even with low-rank approximations, convergence may suffer due to the inherent noise in the estimates, which can delay the stabilization of the approximation.

2. **Lack of Large-Scale Experiments**: While the technique is promising, the experiments are confined to relatively simple models. It would significantly strengthen the paper to include results on larger models, such as transformers or LLMs, to better understand the practical scalability and robustness of the method in high-dimensional regimes.

3. **Missing Utility Guarantees**: The paper does not provide explicit utility bounds or performance guarantees in terms of key DP parameters such as the dimensionality ($d$), dataset size ($n$), and privacy budget ($\epsilon$). Including such an analysis would give a clearer picture of the theoretical trade-offs involved.

---

> ### Author Rebuttal · Authors · 2025-07-31
>
> **We thank the reviewer for their thoughtful and encouraging feedback. We appreciate the recognition of our paper’s clarity, the theoretical depth of our analysis, and the novelty of integrating streaming PCA into adaptive clipping for differentially private optimization. We are especially grateful for the acknowledgment of our method’s strong empirical performance  and the practical significance of addressing gradient geometries in a computationally efficient way.**
>
>  We respond to the specific points raised below.
>
> **Q1) Resource Overhead: A primary limitation of the method is ...**
>
> We thank the reviewer for raising this important point regarding the computational and memory overhead of our method. We acknowledge that storing and updating a full gradient covariance matrix can be expensive in large-scale settings. While much of modern ML focuses on high-dimensional data and large models, tabular datasets have long been, and continue to be, a central part of the ML landscape. These datasets typically have a relatively small number of features, making small models sufficient in many cases. At the same time, tabular data—particularly in domains like healthcare—often involve highly sensitive information, necessitating strict privacy guarantees. High privacy requirements typically limit the number of training iterations and lead to reduced utility. In such settings, our Algorithm 1 (with a full covariance matrix) is well suited: we can compute the full covariance matrix efficiently due to the manageable model size, and GeoClip’s fast convergence enables improved utility in just a few training iterations. We refer the reviewer to Tables 1–3 for results on tabular datasets, Figure 1 (left) for evidence of fast convergence, and Figure 1 (right) for how this fast convergence helps maintain strong privacy.
>
> For larger models, our implementation of Streaming Rank-$k$ PCA maintains only the top-$k$ eigenvectors (out of $d$ total model parameters) and updates them using an $\mathcal{O}(dk^2 + k^3)$ SVD at each step (see the “Low-Rank PCA” section above Section 4). Since $k \ll d$, the cost is dominated by the linear dependence on $d$, making this step efficient even for large models. The memory overhead is also modest. For example, in our USPS experiment (Section 4.4), using a logistic regression model with $d = 2570$ and $k = 100$, Streaming Rank-$k$ PCA only stores a $d \times k$ eigenvector matrix and a length-$k$ eigenvalue vector—far smaller than the full $d \times d$ covariance matrix.
>
> Although GeoClip introduces some computational and memory overhead, we believe this is well justified by the practical benefits. GeoClip converges significantly faster than baseline methods (see Figures 1 and 2), requiring fewer training steps to reach the target accuracy. While each iteration may be slightly more expensive than standard DP-SGD, this is partially offset by the reduced number of steps. Combined with improved privacy—since fewer steps lead to lower total privacy loss—these benefits make the overhead worthwhile.
>
> Regarding the convergence of Streaming Rank-$k$ PCA, we empirically find that GeoClip with Streaming Rank-$k$ PCA maintains strong convergence behavior even for small $k$. As shown in Figure 2, the method still significantly outperforms baseline methods. A direct comparison between the full (Algorithm 1) and low-rank (Algorithm 2) approaches is included in our response to Reviewer 6gkB (Question 6), and we kindly ask the reviewer to refer to it. The results demonstrate that even with small $k$, the performance of Streaming Rank-$k$ PCA remains comparable to that of the full covariance method.
>
> ---
>
> **Q2) Lack of Large-Scale Experiments: While the technique is promising...**
>
> We thank the reviewer for highlighting this important direction. We agree that evaluating GeoClip on larger models is a valuable next step to further demonstrate the scalability of Algorithm 2 with Streaming Rank-$k$ PCA. This work serves as a proof of concept, and the results are encouraging enough to motivate applying the method to larger-scale models.
>
> ---
>
> **Q3) Missing Utility Guarantees: The paper does not provide explicit utility bounds ...**
>
> To relate our convergence bound to DP parameters, we note that the bound in Theorem 1 depends on the noise scale $\sigma$. Abadi et al. [1] show the following relationship between $\sigma$ and the privacy parameters $(\varepsilon, \delta)$, for some constant $c$, number of iterations $T$, and dataset size $n$:
> $$
> \sigma = c \frac{\sqrt{T \log(1/\delta)}}{n \varepsilon}.
> $$
> Substituting this into our convergence bound allows us to express it as a function of $\varepsilon$, $\delta$, and $n$.
>
> Regarding the dependency of our convergence bound on dimensionality, if we simplify the expression in Theorem 1 using the bound on $\beta(a_t)$ from Equation (9) and the optimal transformation derived in Theorem 2 (note that the optimal solution occurs when the constraint in Equation (13) is active, i.e., $\operatorname{Tr}(M_t^\top M_t \Sigma_t) = \gamma$), the convergence bound depends on the dimensionality $d$ only through the term $\sum_{i=1}^d \sqrt{\lambda_{i}}$, where $\lambda_{i}$ is the $i$-th eigenvalue of the gradient covariance matrix. This reflects the effective dimension rather than the ambient dimension. In particular, if $\lambda_{i} = 0$ for some $i$, then that coordinate contributes nothing to the convergence rate. Thus, our bound naturally adapts to the structure of the gradient covariance and degrades gracefully in high-dimensional settings where the spectrum is concentrated in a low-dimensional subspace.
>
> If accepted, we will include this discussion in the final version.
>
>
> [1] Abadi et al., *Deep Learning with Differential Privacy*, in Proceedings of the 2016 ACM SIGSAC Conference on Computer and Communications Security (CCS).
>
> ---
>
> **Q4) Convergence Under Approximation: What is the final convergence rate of GeoClip ...**
>
> We would like to highlight that although the proof of Theorem 1 assumes that $M_t$ is a full-rank $d \times d$ matrix, the same analytical steps extend to the case where $M_t$ is a $k \times d$ matrix with $k < d$, corresponding to a projection onto a lower-dimensional subspace. This is precisely the setting used in streaming rank-$k$ PCA. In this case, Equation (5) becomes:
> $$
> \tilde{g}_t = M_t^+ \tilde{\omega}_t + a_t,
> $$
> where $M_t^+$ denotes the (left) Moore-Penrose pseudoinverse of $M_t$. Consequently, in the convergence bound of Theorem 1, the term $\operatorname{Tr}
> \left[(M_t^\top M_t)^{-1}\right]$ is replaced by $\operatorname{Tr}
> \left[(M_t^\top M_t)^+\right]$, accounting for the lower-rank structure.
>
> We thank the reviewer for highlighting this relevant line of work. While our current analysis does not explicitly assume a low-rank Hessian, our method is naturally aligned with this perspective. Intuitively, the directions where the loss changes most rapidly—i.e., those with high curvature—are also the directions where the gradients tend to vary most across data points. As a result, if the Hessian is approximately low-rank, most of the gradient activity is concentrated in the corresponding low-dimensional subspace. This helps explain why the principal components identified by Streaming PCA tend to align with the top eigenspace of the Hessian. Although we do not yet provide a formal analysis under this assumption, the connection is intuitive, and we see this as an interesting direction for future exploration.
>
> ---
>
> **Q5) Utility of Chosen Preconditioner: In Theorem 2, what is the final utility guarantee ...**
>
> We appreciate the reviewer’s comment. The final convergence bound for the optimal transformation matrix can be obtained by substituting the bound on $\beta(a_t)$ from Equation (9) and the optimal transformation derived in Theorem 2 into the expression in Theorem 1. We note that the optimal solution in Theorem 2 occurs when the constraint in Equation (13) is active, i.e., $\operatorname{Tr}(M_t^\top M_t \Sigma_t) = \gamma$.
>
> The transformation matrix presented in Theorem 2 is chosen to minimize the convergence rate, given a fixed noise scale $\sigma$ determined by the $(\varepsilon, \delta)$-DP budget. While convergence rate is not a direct measure of utility, faster convergence implies that a target utility (e.g., loss or accuracy) can be achieved in fewer iterations. Since privacy loss accumulates over iterations, achieving convergence more quickly results in lower overall privacy cost. Therefore, this choice of preconditioner indirectly improves the privacy-utility trade-off.
>
> ---
>
> **Q6) Faster Preconditioning: Is it possible to prove faster convergence to optimal utility ...**
>
> We thank the reviewer for raising this question. In fact, our analysis in Theorem 1 was motivated by the intuition that faster convergence should enable the algorithm to reach a desired utility level in fewer iterations. While we did not explicitly optimize for the number of iterations, we focused on minimizing the convergence rate, which serves a similar purpose in practice. By accelerating convergence through a well-chosen preconditioner, we reduce the number of iterations required. These two objectives can be seen as closely related approaches to achieving efficient learning.
>
> **We hope this rebuttal effectively addresses the reviewer's concerns.**

---

> > ### Comment · Reviewer_ocmc · 2025-08-02
> >
> > Many thanks to the authors for kindly addressing all my questions—everything is much clearer now! I’m happy with the current score and would like to keep it as is.

---

### Official Review · Reviewer_u6Le · 2025-06-29

**Clarity:** 3
**Significance:** 4
**Originality:** 4
**Rating:** 5
**Confidence:** 4

**Summary:**

This paper observes that if the gradient coordinates are correlated over a datasets, then we should not need to add as much noise during DP training; to leverage this they propose (and note this generalizes past methods) first transforming the gradient, then making it private, then inverting the transform before using SGD. In order to find the best transform they formulate an optimization problem which minimizes the effective amount of noise added while not adding significant error by clipping in the transformed space; this optimization comes from a convergence analysis and minimizing an upper-bound on error. They then provide a closed form solution to the optimization, and further a low-rank approximation for high dimension settings. Algorithmically they use the privatized gradients to estimate the covariance matrix and other terms for the transform, introducing no privacy overhead. All experiments showed improvements over the baselines.

**Questions:**

I could not find some minor experimental details in the main paper or supplementary. Below is a list of some details, and I suggest the authors consider adding a section in the appendix providing more details for reproducability.


1.  How was the lr tuned for experiments in section 4.1?
2. Could authors provide exact details on the synthetic datasets used through the experimental section: e.g., what’s the exact distribution (i.e., how are the features correlated), what was the function for y given x?
3. Could authors add what the grid search for hyperparameters were over for the various experiments, and details on the validation set (if it's not just the training set)
4. Related, how was the batch size chosen in section 4.2 (it seems to be fixed for all methods?)
5. How many trials were done for the experiments in section 4.1 and section 4.4

**Ethical Concerns:**

["NO or VERY MINOR ethics concerns only"]

**Final Justification:**

The strengths from my original review hold, as my only concern was how the hyperparameters and evaluations were run. I saw no issues in these.

As such I maintain my original recommendation of acceptance.

**Limitations:**

Yes

**Quality:**

4

**Strengths And Weaknesses:**

Strengths:

1) Novel and insightful methodology
2) Strong empirical results
3) Well-written

Weaknesses:
1) some minor experimental details seem to be missing, see questions for details

---

> ### Author Rebuttal · Authors · 2025-07-31
>
> **We thank the reviewer for their positive feedback and helpful comments. We are pleased that the reviewer found our methodology to be novel and insightful, appreciated the strong empirical results, and considered the paper well-written.**
>
>  We respond to the reviewer’s comments in detail below.
>
> **I could not find some minor experimental details in the main paper or supplementary. The following is a list of some details, and I suggest the authors consider adding a section in the appendix providing more details for reproducibility.**
>
> We thank the reviewer for pointing this out—we agree that including these details is important for reproducibility. If accepted, we will add a dedicated section in the supplementary material that consolidates all relevant experimental details discussed here and includes additional information to improve clarity and completeness.
>
> ---
>
> **Q1) How was the lr tuned for experiments in section 4.1?**
>
> We tuned the learning rate via grid search over a logarithmic scale ranging from 0.0001 to 10.
>
> ---
>
> **Q2) Could authors provide exact details on the synthetic datasets used through the experimental section: e.g., what’s the exact distribution (i.e., how are the features correlated), what was the function for $y$ given $x$?**
>
> We appreciate the reviewer's comment. The synthetic dataset used in Section 4.1 consists of $n = 20{,}000$ samples and 10 features. To obtain the 5 correlated features, we first generate an $n \times 5$ matrix $Z$ and a $5 \times 5$ matrix $A$, each with entries drawn independently from the standard normal distribution. The correlated features are obtained by multiplying $Z$ by $A$. The remaining 5 features are drawn independently from a standard multivariate normal distribution. The full feature matrix $X$ is constructed by concatenating these two blocks. The target $y$ is generated using a linear function with Gaussian noise: $y = Xw + b + \epsilon$, where $w \sim \mathcal{N}(0, I_{10})$ is a weight vector, $b \sim \mathcal{N}(0, 1)$ is a scalar bias term, and $\epsilon \sim \mathcal{N}(0, 0.01^2)$ is i.i.d. noise.
>
>
> For the synthetic dataset used in Section 4.4, the feature matrix $X$ is constructed in the same way. The logits are computed using the same linear function as before. The probabilities $p$ are then obtained via the sigmoid function, and binary labels are generated as $y = \mathbf{1}(p > 0.5)$  where $\mathbf{1}$ is the indicator function.
>
> If accepted, we will add these details in the final version.
>
> ---
>
> **Q3) Could authors add what the grid search for hyperparameters were over for the various experiments, and details on the validation set (if it's not just the training set)**
>
> We thank the reviewer for the suggestion. In what follows, we explain how we tuned our hyperparameters for all experiments. If accepted, we will also include these details in the paper for completeness.
>
>
> **Parameter $\gamma$:** As mentioned in the experimental results section on page 6, we set the parameter $\gamma$ to $1$, as its effect can be absorbed by tuning $h_2$. Therefore, we only tune $h_2$. We have also provided an explanation of how tuning $\gamma$ would have a similar effect as tuning $h_2$ in our response to Question 4 from Reviewer 6gkB, and we kindly refer the reviewer to that discussion.
>
> **Parameter $h_1$:**  We initially ran experiments using several different $h_1$ values within the range $[10^{-15}, 10^{-6}]$, and observed that our algorithms are robust to changes in $h_1$. Therefore, as stated in Section 4, we set $h_1 = 10^{-15}$ across all experiments.
>
> **Parameter $h_2$:** We first tried $h_2$ values over a logarithmic scale in the range $[1, 1000]$, and observed that $h_2 = 1$ and $10$ perform consistently well across all datasets. Therefore, we select the best of these two values for each dataset in the final experiments.
>
> **Parameters $\beta_1, \beta_2, \beta_3$:**  We again tried several different values of $\beta_1, \beta_2$, and $\beta_3$ over the range $[0.9, 0.999]$, and observed that our algorithms are robust to the choice of these parameters. As stated in Section 4, we used standard values commonly used in optimization, such as $\beta_1 = 0.99, \beta_2 = 0.999$, and $\beta_3 = 0.999$, in all experiments.
>
>
>  **Validation set:** We would like to clarify that we used a separate validation set for tuning hyperparameters and choosing the best model for each dataset. As stated in Section 4, we split each dataset into 80-10-10 train-validation-test sets. For all experiments, we selected the best hyperparameters based on the model's performance on the validation set, trained the best model using different random seeds, and reported the average performance on the test set.
>
> To further explore the questions raised by the reviewer, we also conducted an ablation study on $\gamma$, $h_1$, and $h_2$ using the USPS dataset, under the same settings as in Section 4.4.
> As shown, varying $\gamma$ does not significantly affect performance. The results indicate no noticeable impact from $h_1$, and only minor variations with $h_2$.
>
> **Test Accuracy (%) ( ablation study on $\gamma$ with $h_2 = 10$ and $h_1 = 10^{-15}$)**
>
> | Method                     | γ = 0.2 | γ = 0.6 | γ = 1.0 |
> |---------------------------|--------:|--------:|--------:|
> | GeoClip (Algorithm 2, k = 50)  | 85.537 | 86.182 | 85.752 |
>
>
>
> **Test Accuracy (%)  (ablation study on $h_2$ with $h_1 = 10^{-15}$ and $\gamma = 1$)**
>
> | Method                    | h₂ = 1   | h₂ = 10  | h₂ = 100 | h₂ = 1000 |
> |--------------------------|---------:|---------:|---------:|----------:|
> | GeoClip (Algorithm 2, k = 50) | 86.0215  | 85.7526  | 85.7526  | 85.7526   |
>
>
>
> **Test Accuracy (%)  (ablation study on $h_1$  with $h_2 = 10$ and $\gamma = 1$)**
>
> | Method                    | $h_1 = 10^{-6}$ | $h_1 = 10^{-10}$ | $h_1 = 10^{-15}$ |
> |--------------------------|----------------:|-----------------:|-----------------:|
> | GeoClip (Algorithm 2, k = 50) | 85.752         | 85.752          | 85.752          |
>
> ---
>
> **Q4) Related, how was the batch size chosen in section 4.2 (it seems to be fixed for all methods?)**
>
> We specified the batch size used for each dataset in Section 4.2, within the captions of the tables presenting experimental results: batch size 32 for Diabetes, 64 for Breast Cancer, and 512 for Malware. If accepted, we will make these details more clearly stated in the main text.
>
> ---
>
> **Q5) How many trials were done for the experiments in section 4.1 and section 4.4?**
> For the experiments in Section 4.1 and Section 4.4, we selected the best hyperparameters based on the model's performance on the validation set, trained the best model using $20$ different seeds, and reported the average performance on the test set. If the reviewer meant something else by ``trials", we humbly ask them to clarify it.
>
> **We hope our responses adequately address the reviewer's concerns.**

---

> > ### Comment · Reviewer_u6Le · 2025-08-03
> >
> > Yes this helped clarify where the hyperparameters came from and the overall statistical results!  I will keep my score.

---

### Official Review · Reviewer_6gkB · 2025-06-30

**Clarity:** 3
**Significance:** 2
**Originality:** 2
**Rating:** 4
**Confidence:** 4

**Summary:**

The paper proposes a method to improve the utility of differentially private stochastic gradient descent (DP-SGD) by transforming gradients into an appropriate coordinate system where clipping can be performed more effectively. Theoretical analysis is provided to justify the approach, and experimental results on synthetic data and fine-tuning tasks demonstrate its potential. However, its applicability to larger model training is required.

**Questions:**

See weaknesses

**Ethical Concerns:**

["NO or VERY MINOR ethics concerns only"]

**Final Justification:**

I still have concerns about the practical applicability of the method, but the paper presents reasonable theoretical results, and therefore I am increasing my rating.

**Limitations:**

Yes

**Quality:**

3

**Strengths And Weaknesses:**

Strengths:
- The paper tackles an important challenge in differentially private stochastic gradient descent (DP-SGD), namely improving gradient clipping, which is a critical bottleneck for utility.
- The proposed method (transforming gradients to a more appropriate coordinate system) is clean and intuitive. The theoretical exposition and writing are clear and easy to follow.

Weaknesses:
1. While the idea of coordinate transformation is novel in presentation, the resulting theoretical insights appear closely related to prior work such as AdaCLIP, potentially limiting the originality of the contribution.

2. A key concern is the limited scope of empirical validation. Unlike AdaCLIP, which demonstrated results on MNIST and CelebA using CNNs and ResNets, this paper restricts experiments to synthetic data or simple final-layer fine-tuning. This raises questions about practical utility in the settings where DP-SGD is most challenged (low performance in practical models).

3. Related works that project gradients into lower-dimensional subspaces for utility improvement (see [1, 2]) are not discussed or compared. Including them would provide important context and highlight the positioning of this method within the broader landscape.
Especially with Low-Rank PCA results (Algorithm 2).

4. The choice of the parameter $\gamma$ remains unclear, even when combined with $h_2$.

5. A more principled or empirical justification of $h_1$ and $h_2$ is necessary.

6. If many eigenvalues of the covariance matrix are near zero, it raises the question of why the method does not project gradients onto the effective low-dimensional subspace directly, which could further reduce noise and improve utility. This question requires the comparison between Algorithm 1 and Algorithm 2.

[1] Bu, Zhiqi, et al. "Fast and memory efficient differentially private-sgd via jl projections." NeurIPS 2021
[2] Zhou, Yingxue, Steven Wu, and Arindam Banerjee. "Bypassing the Ambient Dimension: Private SGD with Gradient Subspace Identification." ICLR 2021

---

> ### Author Rebuttal · Authors · 2025-07-31
>
> **We thank the reviewer for their valuable feedback. We are glad that the paper’s focus on improving gradient clipping in DP-SGD was recognized as important, and that our proposed coordinate transformation was seen as clean and intuitive. We also appreciate the positive feedback on the clarity of our theory and writing.**
>
>  Below, we address the reviewer’s remaining comments.
>
> **Q1) While the idea of coordinate ...**
>
> We appreciate the reviewer’s comment. While our method can be seen as a generalization of AdaClip—as discussed in the paper—we emphasize that this extension is non-trivial, both conceptually and theoretically. Our analysis handles the complexity introduced by the basis transformation, going beyond AdaClip's coordinate-wise scaling.
>
> Whereas AdaClip constructs clipping thresholds using only the coordinate-wise mean and variance of the gradients, our method estimates a structured transformation based on the eigen-decomposition of the gradient covariance matrix. Estimating eigenvalues and eigenvectors is a more sophisticated approach which then assures significantly better results.
> To this end, we designed an efficient streaming algorithm to estimate the top eigenvalues and eigenvectors in a scalable manner. Moreover, to the best of our knowledge, this is the first work to combine two lines of research: adaptive clipping and projection onto gradient subspace representations. We hope that our contributions in these directions are recognized as distinct and meaningful.
>
> ---
>
> **Q2) A key concern is the limited scope of empirical validation...**
>
> We appreciate the reviewer’s comment. To address this, we first clarify the experimental setups used in AdaClip [1] and in our work, and then highlight a challenging setting where GeoClip shows a clear advantage over existing baselines.
>
> The AdaClip paper evaluated a synthetic regression task and MNIST, using only logistic regression and a one-hidden-layer neural network. For logistic regression, the full 784-dimensional MNIST input was used. For the neural network, the input was first reduced to 60 dimensions using differentially private PCA, followed by training a fully connected layer with 1000 hidden units. The privacy budget was split between the PCA step and model training. As far as we can determine from the cited paper, no CNNs or other deep architectures such as ResNets were evaluated, and MNIST was the only real-world dataset considered.
>
> GeoClip with the full covariance estimator (Algorithm 1) was evaluated on synthetic, tabular, and image datasets. For tabular data, we used logistic or linear regression on Diabetes (10 features), Breast Cancer (30 features), and Android Malware (241 features). We also transferred a convolutional network pretrained on MNIST to Fashion-MNIST and fine-tuned the final fully connected layer under DP. This setting involved 510 trainable parameters. For higher-dimensional cases, Algorithm 2 (with streaming rank-$k$ PCA) was evaluated on a synthetic dataset and on USPS (256 features) using multiclass logistic regression (2570 parameters).
>
> Regarding the practicality of GeoClip in challenging settings, we offer the following clarification. While much of modern ML focuses on high-dimensional data and large models, tabular datasets have long been, and continue to be, a central part of the ML landscape. These datasets typically have a relatively small number of features, making small models sufficient in many cases. At the same time, tabular data—particularly in domains like healthcare—often involve highly sensitive information, necessitating strict privacy guarantees. High privacy requirements typically limit the number of training iterations and lead to reduced utility. In such settings, Algorithm 1 is well suited: it allows for full covariance computation and benefits from GeoClip’s fast convergence, enabling improved utility in just a few training steps. We refer the reviewer to Tables 1–3 (tabular datasets) and Figure 1 (fast convergence) for supporting evidence.
>
> For larger models and datasets, Algorithm 2 addresses similar challenges under strong privacy constraints. Our proposed streaming rank-$k$ PCA enables faster convergence, allowing a given level of utility to be achieved in fewer iterations. Please see Figure 2 in the main paper, as well as the privacy curves in Figure 4 of the supplementary material, which illustrate how fast convergence supports stronger privacy guarantees. We hope these helps address the reviewer’s concerns about the practicality of GeoClip.
>
> We acknowledge that more experiments on large-scale settings would further support the scalability claims of Algorithm 2. This work serves as a proof of concept, and the results are encouraging enough to motivate applying the method to larger models.
>
> [1] Pichapati, Venkatadheeraj, et al. ``AdaClip: Adaptive Clipping for Private SGD.'' arXiv preprint arXiv:1908.07643 (2019).
>
> ---
>
> **Q3) Related works that project gradients into lower-dimensional subspaces for utility improvement (see [1, 2]) ...**
>
> We appreciate the references provided by the reviewer. In what follows, we discuss the key differences.
>
> A key difference between our approach and DP-SGD-JL [Bu et al., 2021] lies in the objective. DP-SGD-JL aims to improve computational efficiency—specifically memory and speed—by approximating per-sample gradient norms using Johnson–Lindenstrauss projections, while maintaining similar utility to standard DP-SGD. In contrast, our method is designed to improve the privacy–utility tradeoff.
>
> We also differ from the approach in [Zhou et al., 2021], which assumes access to a public dataset drawn from the same distribution as the private data in order to estimate a low-dimensional gradient subspace, thereby avoiding additional privacy cost. Obtaining such a representative public dataset may not always be practical, and their analysis shows that the quality of the estimated subspace—and thus the effectiveness of the projection—depends on the size of the public data. In contrast, our method does not require a public dataset and incurs no additional privacy cost for identifying the subspace. Instead, we estimate the gradient geometry by reusing previously released noisy gradients.
>
> If accepted, we will incorporate these comparisons into the paper.
>
> ---
>
> **Q4) The choice of the parameter $\gamma$ remains unclear, even when combined with $h_2$**
>
> In our formulation, $\gamma$ bounds the probability that a gradient is clipped (see the discussion above Equation (13)). The parameter $h_2$ also affects the clipping probability by influencing the clamped eigenvalues and consequently, the transformation matrix $M$.  To simplify hyperparameter tuning, we set $\gamma = 1$ and tuned only $h_2$, since both parameters have similar effects.
>
> We conducted an ablation study on $\gamma$ in our response to Question 3 from reviewer u6Le, and we kindly refer the reviewer to that discussion due to space limitations. If accepted, we will include a more extensive ablation study varying both $\gamma$ and $h_2$ independently.
>
> ---
>
> **Q5) A more principled or empirical justification of $h_1$ and $h_2$  is necessary.**
>
> We thank the reviewer for this comment. As noted on page 5, we introduce $h_1$ and $h_2$ to stabilize the transformation by avoiding issues from near-zero eigenvalues and limiting the influence of large, noisy ones. We had such ablation studies performed but missed adding them to the paper. We included them in our response to Question 3 from reviewer u6Le, and we kindly refer the reviewer to that discussion due to space limitations. The results show no impact from varying $h_1$, and only minor changes with $h_2$. We also refer the reviewer to our earlier response regarding the effect of $h_2$ on clipping.
>
> ---
>
> **Q6) If many eigenvalues of the covariance matrix are near zero...**
>
> We emphasize that both of our algorithms adapt noise and clipping to the geometry of the gradient distribution. Algorithm 2 (streaming rank-$k$ PCA) projects gradients from the $d$-dimensional space onto a $k$-dimensional subspace aligned with the top-$k$ eigenvectors of the gradient covariance. In contrast, Algorithm 1 (with full covariance) transforms gradients into the full eigenbasis and retains all $d$ directions. Crucially, noise is scaled by the corresponding eigenvalue, so directions with near-zero eigenvalues receive little to no noise. This makes Algorithm 1 a soft generalization of Algorithm 2: it retains all directions while still emphasizing dominant ones and is at least as expressive.
>
> To support this, we compare the accuracy of GeoClip using Algorithm 1 and Algorithm 2 with $k=50$ on the same synthetic dataset from Section 4.4. As shown in Figure 2 of our paper, Algorithm 2 already outperforms other baselines. In the table below, we show that even with $k=50$, Algorithm 2 achieves comparable accuracy to Algorithm 1, which computes the full covariance. All results are averaged over 20 random seeds.
>
> | Method                    | Iter 10       | Iter 30       | Iter 50       | Iter 70       |
> |--------------------------|---------------|---------------|---------------|---------------|
> | GeoClip ( Algorithm 2, k = 50) | 85.51 ± 0.83 | 87.93 ± 0.52 | 88.02 ± 0.60 | 88.11 ± 0.67 |
> | GeoClip  (Algorithm 1)                | 87.34 ± 0.75 | 89.09 ± 0.48 | 89.24 ± 0.40 | 89.26 ± 0.42 |
>
> If accepted, these comparisons along with the corresponding plots, will be provided in the final version to further illustrate their relationship.
>
> **We hope these responses are satisfactory and convincing to the reviewer.**

---

### Official Review · Reviewer_Y5Ye · 2025-07-01

**Clarity:** 4
**Significance:** 3
**Originality:** 2
**Rating:** 5
**Confidence:** 3

**Summary:**

This paper introduces a new method called GeoCLIP to obtain better privacy/utility tradeoffs in the DP-SGD algorithm. This is an improvement over Adapclip, which is a baseline they compare against.

Their hypothesis is the following: noise + clipping operations operate on the coordinates disjointly, failing to recognize situations in which they are correlated. They may yield to an excessive amount of noise added and may degrade utility. Therefore, the authors propose to apply an affine transformation to the gradients before the noise+clipping operation and to invert this transformation after to project back into the gradient space.  The goal of this affine transformation is to decorrelate the coordinates of the gradients, to allow DP-SGD to operate in a regime it is the most comfortable with.

More precisely:
- the gradient is transformed as $M(g-a)$
- it is clipped and noisified: $\text{clip}(g)+\text{noise}$
- the gradient is projected back $M^{-1}g+a$

An analysis of the error introduced is performed. Several terms are identified:
- optimization gap
- gradient variance term
- noise injection term
- clipping error term
This allows the optimization of each term separately, by solving the corresponding optimization program. Authors found something akin to whitening transformation (but not quite the same), and showed that their version outperforms whitenening in this scenario.

Moreover, since the optimal form relies on mean/covariance statistics of gradients (which are private), the authors propose to rely on an exponential moving average. Finally, storing the full covariance matrix is prohibitively expensive, so the author proposes to rely on a new algorithm called *Streaming Rank-kPCA* to update it timestep after timestep.

The evaluation is performed on:
- synthetic data, in which some coordinates are artificially correlated
- tabular data (Android Malware, Diabetes, Breast Cancer) in low dimension and small sample size
- USPS dataset on images

The baselines are DP-SGD, AdapClip, and a method based on quantiles.
The privacy is computed using *Connect-the-Dots* accountant.

**Questions:**

### Decorrelation sanity check

If you modify the experiment of Section 4.1 so that *all the features are decorrelated*, do you observe a difference between your method, Adaclip, DP-SGD and quantiles? Which one performs best here?

### Quantile-based clipping

Does it make sense to combine your method with *quantile-based clipping*? Do you expect to witness an improvement?

### Full covariance

Your method essentially leverages the full covariance matrix, even though you propose a way to relax this constraint. Are you aware of other privacy-preserving algorithms in the literature that leverage the full covariance, and could benefit from your *Streaming Rank-kPCA* trick?

**Ethical Concerns:**

["NO or VERY MINOR ethics concerns only"]

**Final Justification:**

I found the rebuttal very satisfactory on the central contribution of the paper: the motivation behind correlated gradients that pop up during training. Authors did a good job at providing an additional sanity check, and pedagogical examples.

**Limitations:**

The authors discussed all the main limitations:
* "**additional testing is needed to see how our algorithm performs in more generality**"
* "The linear transformation involved in GeoClip requires an additional computation via an eigendecomposition"
* "The algorithm introduces additional hyperparameters (most notably h2) compared to standard DP-SGD, which must be tuned for optimal performance"

**Paper Formatting Concerns:**

No concerns.

**Quality:**

3

**Strengths And Weaknesses:**

## Strength

### Principled method

The method is well-motivated. It follows the strategy of Adaclip: decompose the error in terms when applying a transformation to the gradients. Formulating the optimal choice of affine transformation as an optimization problem is clever. The extension to situations in which the full covariance is not available is also interesting. The method looks sound (I did not check the proofs).

### Convincing small-scale experiments

The experiments displayed here are very convincing, showing a clear improvement over AdaClip and DP-SGD baselines, on small-scale datasets.

## Weakness

### Unsufficient evaluation at medium and large scale

**Small datasets**. The biggest dataset is USPS which is: (i) easy (ii) small and low dimensional. It is unclear if the benefits observed would transfer to bigger datasets that are higher dimensional.

**Streaming Rank-kPCA**: as it stands, a benchmark is lacking on the memory usage/speed of the algorithm. How big can the network be on a standard GPU, before *Streaming Rank-kPCA* becomes a bottleneck? How does that compare to competitors?

**Decorreleted basis.** The motivation of the work - correlated coordinates in gradients - is believable for some highly structured datasets like USPS, which have patch of pixels that tend to look the same, at the same places. However, for datasets like CIFAR-10 or TinyImageNet, images are more decorrelated; and so maybe are the gradients too!

**LLM**. It could be interesting to take a small LLM (e.g. transformer with <40M parameters, small vocabulary, autoregressive training) on a small text dataset (any subset from The Pile would do, for example) and see how your method compares against DP-SGD on it.

### Missing baselines and related work

Adapclip is not the only improvement over DP-SGD for privacy training. Some other papers could be discussed or compared against:

* Lee, Jaewoo, and Daniel Kifer. "Scaling up differentially private deep learning with fast per-example gradient clipping." arXiv preprint arXiv:2009.03106 (2020).
* Li, Xuechen, Florian Tramer, Percy Liang, and Tatsunori Hashimoto. "Large Language Models Can Be Strong Differentially Private Learners." In International Conference on Learning Representations (2022).
* Bu, Zhiqi, Jialin Mao, and Shiyun Xu. "Scalable and efficient training of large convolutional neural networks with differential privacy." Advances in Neural Information Processing Systems 35 (2022): 38305-38318.
* Bu, Zhiqi, Yu-Xiang Wang, Sheng Zha, and George Karypis. "Differentially private optimization on large model at small cost." In International Conference on Machine Learning, pp. 3192-3218. PMLR, 2023.
* Béthune, Louis, Thomas Massena, Thibaut Boissin, Aurélien Bellet, Franck Mamalet, Yannick Prudent, Corentin Friedrich, Mathieu Serrurier, and David Vigouroux. "DP-SGD Without Clipping: The Lipschitz Neural Network Way." In The Twelfth International Conference on Learning Representations (2024).

# Recommendation

Overall, the paper looks correct and improves the DP-SGD and AdapClip by considering a full rank affine transformation, instead of a diagonal rescaling. Extending Adapclip to the full rank setup is non-trivial when considering the technical difficulties implied (e.g. covariance estimation, efficient storage, solving the optimization problem to find the"withening"), so this work *is not* incremental.  The algorithm is more tedious to implement (necessitating the use of online PCA) but this yields significant improvements over the competitor, on small-scale benchmarks.

That being said, it is unclear how the method performs beyond this small scale. If authors can do a commanding job of demonstrating feasability at larger scale for the rebuttal, I'd be happy to increase my score even more. In any case, the paper is interesting and I am leaning toward acceptance.

---

> ### Author Rebuttal · Authors · 2025-07-30
>
> **We thank the reviewer for the constructive and detailed feedback. The recognition of the principled motivation behind the method and the acknowledgment that extending Adaclip to the full-rank setting is a non-trivial and meaningful contribution are appreciated. It is encouraging that the resulting improvements were found to be significant, and that the experiments on small-scale datasets were seen as very convincing, demonstrating clear gains over the baselines.**
>
> We address the reviewer’s concerns below.
>
> **Q1) Small dataset**
>
> We appreciate the reviewer’s comment. While much of modern ML focuses on high-dimensional data, tabular datasets have long been, and continue to be, a central part of ML landscape. These datasets typically have a relatively small number of features, making small models sufficient in many cases. At the same time, tabular data—particularly in healthcare—often involve highly sensitive information, necessitating strict privacy. High privacy requirements limit the number of training iterations and lead to reduced utility. In such settings,  Algorithm 1 is well suited: it allows to compute the full covariance matrix efficiently due to the manageable model size, and its fast convergence enables improved utility in just a few iterations.
>
> We kindly refer the reviewer to our response to Question 6 from reviewer 6gkB, where we compare our two algorithms and show that even with a small $k$, the performance of Streaming Rank-$k$ PCA (Algorithm 2) remains comparable to that of Algorithm 1, which uses the full covariance matrix. This indicates that Algorithm 2 is effective under resource constraints.
>
>  We acknowledge that more experiments on large-scale settings would strengthen the scalability claims of Algorithm 2. This work serves as a proof of concept, and the results are encouraging enough to motivate applying the method to larger models.
>
> ---
>
> **Q2) Streaming Rank-k PCA**
>
> Our implementation maintains only the top-$k$ eigenvectors (out of $d$ total model parameters) and updates them using an $\mathcal{O}(dk^2 + k^3)$ SVD at each step (see the “Low-Rank PCA” section above Section 4). Since $k \ll d$, the cost is dominated by the linear dependence on $d$, making this step efficient even for large models if $k$ is small.
>
> The memory overhead is also modest. For example, in USPS experiment (Section 4.4), using a logistic regression model with $d = 2570$ and $k = 100$, Streaming Rank-$k$ PCA only stores a $d \times k$ eigenvector matrix and a length-$k$ eigenvalue vector—far smaller than the full $d \times d$ covariance matrix.
>
> Although it introduces some overhead, we believe it is well-justified by the practical gains. It converges significantly faster than baseline methods (see Figure 2), requiring fewer training iterations to reach target accuracy. While each iteration may be slightly more expensive than DP-SGD, this is partially offset by the reduced number of steps. Combined with improved privacy—since fewer steps lead to lower total privacy loss—these benefits make the overhead worthwhile.
>
> If accepted, we will conduct more extensive benchmarking to compare the computation time and memory usage of GeoClip to the other versions of DP-SGD.
>
> ---
>
> **Q3) Decorreleted basis**
>
> While there may be a more direct relationship between input feature correlations and gradient coordinate correlations in linear models, this connection does not generally hold in nonlinear architectures. Nonlinear activations, weight sharing, and depth introduce interactions that can create dependencies between gradient coordinates—even when the input features are uncorrelated.
>
> Consider a simple MLP with two independent inputs $x_1$ and $x_2$, a hidden layer computing
> $h = \mathrm{ReLU}(w_1 x_1 + w_2 x_2)$,
> and output
> $y = \nu \cdot h$,
> where $\nu$ is the output-layer weight. With squared error loss, the gradients are:
>
> $$
> \frac{\partial L}{\partial w_1} = \alpha(x_1, x_2) \cdot x_1, \quad
> \frac{\partial L}{\partial w_2} = \alpha(x_1, x_2) \cdot x_2,
> $$
>
> where
>
> $$
> \alpha(x_1, x_2) = \left( \nu \cdot \mathrm{ReLU}(w_1 x_1 + w_2 x_2) - y_{\text{true}} \right) \cdot \nu \cdot \mathbf{1}_{w_1 x_1 + w_2 x_2 > 0}.
> $$
>
> Although $x_1$ and $x_2$ are uncorrelated, both gradient components are scaled by the same nonlinear factor $\alpha(x_1, x_2)$, introducing dependency. These effects are amplified in deeper networks. Prior work shows that gradients often concentrate in low-dimensional subspaces [1], resulting in geometry that GeoClip is designed to exploit.
>
> [1] Gur-Ari et al. Gradient Descent Happens in a Tiny Subspace.
>
> ---
>
> **Q4) LLM**
>
> We agree that evaluating the method on small LLMs would be a valuable direction to explore. This work was intended as a proof of concept, and our focus was on demonstrating the benefits of the proposed method in controlled settings. We plan to extend our experiments to include such models, particularly in the context of fine-tuning LLMs on sensitive datasets.
>
> ---
>
> **Q5) Missing baselines and related work**
>
> We appreciate the wide-ranging references provided by the reviewer. Our primary goal in comparing against existing variations of DP-SGD was to focus specifically on methods related to clipping. We acknowledge that some of the references pointed out by the reviewer make improvements to DP-SGD, but in aspects other than the clipping. For example, the first four references [Lee \& Kifer (2020), Li et al. (2022), Bu et al. (2022), Bu et al. (2023)] focus on practical aspects such as improving speed, reducing memory usage, and lowering computational cost—directions that are orthogonal to our focus on the privacy-utility trade-off. On a different note, Béthune et al. (2024) takes a different approach by avoiding clipping altogether through Lipschitz-constrained networks, which is also complementary to our method. In the interest of maintaining focus, we elected not to include comparisons to these methods.
>
> ---
>
> **Q6) Decorrelation sanity check**
>
> The experiment in Section 4.1 involves linear regression on a synthetic Gaussian dataset with 10 features (five correlated). As suggested, we repeated the experiment with the same setup but with all 10 features drawn independently. We have collated our results in the Tables below.
>
> In linear regression with uncorrelated features, gradient coordinates tend to become uncorrelated as the model is learned. Let
>
> $x \sim \mathcal{N}(0, I_d)$
>
> and
>
> $y = (w^*)^\top x + \epsilon$
>
> where $\epsilon$
> is zero-mean noise with variance $\sigma^2$, independent of $x$. The gradient is
> $
> \nabla_w L = (w^\top x - y) \cdot x = e \cdot x,
> $
> where
> $
> e = w^\top x - y = (w - w^*)^\top x - \epsilon.
> $
>
> As training progresses and  $w \to w^*$, the term
>
> $(w - w^*)^\top x$
>
> vanishes, and the residual becomes $e \approx -\epsilon$, which is independent of $x$. In this regime, the gradient simplifies to $\nabla_w L \approx -\epsilon x$, and the gradient covariance becomes
> $$
> \mathbb{E}[\nabla_w L (\nabla_w L)^\top] \approx \mathbb{E}[\epsilon^2 x x^\top] = \mathbb{E}[\epsilon^2] \cdot \mathbb{E}[x x^\top] = \sigma^2 I_d.
> $$
> This reflects uncorrelated gradient coordinates. This structure becomes more accurate later in training; during early iterations, the residual $e$ may still depend on $x$, and the gradient coordinates can be correlated. Thus, GeoClip tends to outperform AdaClip in the early stages of training (where only a small portion of the privacy budget has been consumed and the model operates in a higher privacy regime), and behaves similarly as the model converges. GeoClip consistently outperforms the baseline, as shown in Table 1.
>
> **Table 1 (Test MSE over 10 seeds)**
> | Method    | Epoch 1           | Epoch 3           | Epoch 5           | Epoch 7           |
> |-----------|--------------------|--------------------|--------------------|--------------------|
> | GeoClip   | 2.22 ± 0.34   | 0.00027 ± 0.00004| 0.0001 ± 0.000002| 0.0001 ± 0.000003|
> | AdaClip   | 2.52 ± 0.30     | 0.0087 ± 0.005    | 0.003 ± 0.001   | 0.003 ± 0.002    |
> | DP-SGD    | 6.96 ± 0.69     | 4.04 ± 0.54     | 1.91 ± 0.37     | 0.58 ± 0.20     |
> | Quantile  | 8.46 ± 0.77   | 4.04 ± 0.47   | 0.11 ± 0.02   | 0.0006 ± 0.0002|
>
> We also conducted an additional experiment using the synthetic dataset from Section 4.4 (400 Gaussian features with logistic regression), this time with all features drawn independently. This illustrates how the nonlinearity induced by a sigmoid can introduce gradient correlations even under feature independence—structure that GeoClip is able to exploit. The results, shown in Table 2, align with our response to the reviewer’s earlier comment on the decorrelated basis.
>
> **Table 2 (%) (Test Accuracy over 10 seeds)**
> | Method               | Iter 10         | Iter 30         | Iter 50         | Iter 70         |
> |----------------------|------------------|------------------|------------------|------------------|
> | GeoClip ($k=50$) | 81.26 ± 1.20     | 84.03 ± 0.68     | 83.92 ± 0.77     | 83.51 ± 0.53     |
> | AdaClip              | 53.95 ± 2.90     | 65.06 ± 2.25     | 74.07 ± 1.48     | 79.50 ± 1.23     |
> | DP-SGD               | 54.67 ± 2.88     | 62.74 ± 2.53     | 69.05 ± 1.89     | 73.85 ± 1.67     |
> | Quantile             | 49.77 ± 1.67     | 51.26 ± 1.75     | 57.31 ± 1.74     | 68.34 ± 1.27     |
>
> ---
>
> **Q7) Quantile-based clipping**
>
> Combining with quantile-based clipping is unlikely to help, as our method already derives an optimal transformation matrix that includes both a change of basis and clipping.
>
> ***
> **Q8) Full covariance**
>
> At this point, we are not aware of differentially private algorithms that explicitly use the full gradient covariance; however, there is a growing interest in enhancing Adam-like optimizers with methods that require full gradient covariance matrices. Such approaches, with or without privacy could directly benefit from our Streaming Rank-$k$ PCA.
>
> **We hope these responses convincingly address the reviewer's concerns.**

---

> > ### Author Response · Authors · 2025-08-05
> >
> > We thank the reviewer again for the detailed comments and feedback. We have submitted a detailed rebuttal addressing the points they raised, and we would be grateful if they could let us know whether their concerns have been resolved. We are happy to clarify any part that remains unclear.

---

> > > ### Comment · Reviewer_Y5Ye · 2025-08-05
> > >
> > > I’d like to thank the authors for their detailed answer. I find the explanations on “decorrelated basis” and “decorelation sanity check” very convincing. In addition, the analysis of complexity of streaming rank k PCA convinces me of the practicality of the method.
> > >
> > > I have no further concerns and I increased my score.

---

### Public Comment · ~Marc_Molina_Van_den_bosch1 · 2025-11-17
**Potential privacy bug in reference implementation: repeated RNG reseeding, and reused noise**

Dear authors,

While going through your implementation to understand the GeoClip mechanism, I found a major issue that directly affects the validity of your privacy claims. In the noise-generation routine, the global RNG is reset on *every* call (`torch.manual_seed(..)` / `torch.cuda.manual_seed(..)` inside the per-batch function, geoclip.py line 42, in geoclip_rank_k.py there is the exact same bug line 49). This causes the Gaussian noise added at each step to be the **same deterministic vector**, not independent samples. As a result, the mechanism is not differentially private as stated, because the noise is not renewed, hence the network just learns to compensate this constant perturbation.

After removing the per-step reseeding to restore the intended independent noise draws, I observed that training actually **diverges** on the datasets I tested (including your own). This strongly suggests that a substantial part of the reported stability and performance is coming from this deterministic noise artifact rather than from the GeoClip method itself. To check this behavior it would have been a very simple check to just print the noise you add at each step and you would have seen how is fixed for each step.

I’m pointing this out because it is not a small bug, it fundamentally undermines both the privacy guarantee and the empirical results. I cannot determine whether this was an oversight or intentional, but in its current form the implementation does not match the mechanism described in the paper, and correcting the issue leads to completely different behavior.

Best regards.

---

> ### Public Comment · ~Atefeh_Gilani1 · 2026-02-07
> **Response to the comment regarding reseeding**
>
> We apologize for the mistake in the GitHub release. During code transfer and reorganization, a few unintended changes were introduced into the public version, which led to the divergent behavior the commenter observed. Unfortunately, this behavior only appeared after many training iterations, which is why it was not detected in the short demo runs we provided on GitHub. The per-step reseeding was another overlooked change added only in the GitHub version to make the demo runs exactly reproducible.
>
> We emphasize that the original code used to obtain the paper’s results did not include any of these issues and has now been pushed to GitHub. Importantly, the divergence is unrelated to the reseeding itself. We explicitly verified that in the original GitHub snapshot, the same divergence was observed even when the per-step reseeding was not removed, showing that the behavior cannot be attributed to the commenter's claim. Consequently, the conclusion that the reported stability and performance rely on the reseeding does not follow.

---

### Note · Authors · 2025-08-15

We sincerely thank the reviewers and Area Chair for their time, effort, and constructive feedback.

Our paper proposes GeoClip, a geometry-aware approach to differentially private SGD that leverages the structure of the gradient distribution to improve the utility–privacy tradeoff. To ground this in theory, we provide a convergence theorem showing how the choice of basis plays a key role in convergence, and formulate a convex optimization problem—along with a closed-form solution—to compute the optimal transformation that accelerates convergence. This faster convergence enables the model to reach target utility in fewer iterations, thereby consuming less privacy budget and improving the utility-privacy tradeoff. As a result, GeoClip achieves high utility even in challenging high-privacy settings where only a limited number of training iterations can be afforded.

The optimal transformation depends on the gradient covariance, and we develop two practical algorithms to estimate it with no additional privacy cost—both rely only on previously released noisy gradients. One uses the full covariance matrix and is suitable for smaller datasets, such as tabular data in the healthcare domain; the other employs a streaming low-rank approximation and scales to large models. Finally, our experiments on synthetic, tabular, and image datasets validate the theory, showing consistent improvements in the utility–privacy tradeoff enabled by faster convergence.

We are pleased that all reviewers engaged with us during the discussion period and recommended acceptance. We are grateful for their thoughtful input, which helped improve the paper, and we will incorporate their suggestions into the final version.

---

### Decision · Program_Chairs · 2025-09-17

**Decision:**

Accept (poster)

**Comment:**

This paper proposes a theoretically principled algorithm to improve differentially private machine learning from the perspective of minimizing the upper bound of the convergence error. One of the closest prior works is AdaClip, which this work generalizes by leveraging the full covariance matrix as opposed to a diagonal matrix. It develops methods to estimate the covariance matrix without incurring extra privacy loss. Though on some datasets empirical improvements are not significant and it might be difficult to scale to large-scale settings (due to the decomposition of large matrices), I think there are many merits in the proposed theory and methods as mentioned by the reviewers, hence recommend acceptance.